# Reward-Preserving Counterfactual State Editing for Offline Reinforcement Learning

**Siyu Wang**[1]   **Xiaocong Chen**[2]   **Mingming Gong**[3]   **Yong Li**[4]   **Quan Z. Sheng**[1]   **Lina Yao**[5]

## Abstract

Transformer sequence models such as Decision Transformer can learn strong offline policies from logged trajectories, but they often suffer from causal confusion: reliance on spurious correlations that predict reward in the data but do not reflect the true causal mechanisms of the environment. We propose CSET (Counterfactual State Editing Transformer), which improves robustness in strictly offline reinforcement learning without learning environment transition dynamics. On the data side, CSET fits a causal reward model as a conditional variational autoencoder and uses a counterfactual state generator to propose minimally edited observations whose predicted reward matches the factual reward, under a normalized move-band constraint and an acceptance gate that enforce plausibility and reward consistency; augmentation replaces only the observation token to avoid synthetic successor transitions. On the model side, CSET uses a causally structured hybrid transformer: modality-specific convolutional encoders process return-to-go, state, and action streams, and a final attention block is softly supervised so action prediction focuses on its direct causal parents. Experiments on D4RL locomotion, AntMaze, and offline recommendation benchmarks show consistent gains within the DT family, and CSET remains substantially more robust than strong value-based and DT baselines under injected spurious distractors.

## 1. Introduction

The success of large sequence models, particularly Transformers, in natural language processing (Devlin et al., 2019; Brown et al., 2020; Wolf et al., 2020) and computer vision (Chen et al., 2020; Ramesh et al., 2021; Reed et al., 2022) has motivated their use for sequential decision making. In offline reinforcement learning (RL), Decision Transformer (DT) (Chen et al., 2021) shows that policy learning can be cast as conditional sequence modeling: given a context of returns-to-go, states, and past actions from a fixed dataset, the model predicts the next action to achieve a desired objective. Importantly, DT-style methods are *model-free*: they learn a policy directly from logged trajectories without learning an explicit transition model or interacting with the environment.

However, decision-making data are not like text or images. Trajectories are generated by an environment with Markovian transition structure and causal mechanisms, and offline datasets often contain strong correlations that are not stable under intervention. As a result, Transformer-based offline policies can suffer from *causal confusion* (Lyle et al., 2021; Urpí et al., 2024): they exploit spurious features that predict reward in the dataset but are not causally related to reward in the environment (Gupta et al., 2023; Tien et al., 2022). For example, a policy may latch onto a background feature that happens to co-occur with high reward in the log, leading to brittle behavior when that feature changes at deployment. This risk is amplified by the flexibility of global self-attention: without task-specific inductive bias, attention can focus on any predictive token, including non-causal shortcuts (Agarwal et al., 2023; Kim et al., 2024). Since offline RL cannot collect new data to break these correlations, improving robustness and generalization requires injecting causal structure rather than relying on correlation alone (Richens & Everitt, 2024).

A natural response is to learn a dynamics model and generate counterfactual rollouts (Yu et al., 2020). However, in strictly offline RL, this is difficult to do reliably: coverage is limited, model errors are hard to validate outside the dataset support, and multi-step rollouts can compound errors (Sims et al., 2024). Moreover, DT-based methods do not require a learned transition model (Chen et al., 2021), and offline

---

[1]School of Computing, Macquarie University, Australia [2]CSIRO's Data 61, Australia [3]School of mathematics and statistics, University of Melbourne, Australia [4]Department of Electronic Engineering, Tsinghua University, China [5]School of Computer Science and Engineering, University of New South Wales, Australia. Correspondence to: Siyu Wang <siyu.wang@mq.edu.au>.

*Proceedings of the 43rd International Conference on Machine Learning*, Seoul, South Korea. PMLR 306, 2026. Copyright 2026 by the author(s).

benchmarks such as D4RL provide fixed logged transitions rather than an interactive environment for interventions (Fu et al., 2020). These constraints motivate a different goal: instead of enforcing full transition validity under a learned dynamics model, we focus on forms of causal consistency that are checkable from offline data and directly useful for training model-free sequence policies.

We argue that reducing causal confusion requires strengthening both the data and the model. On the data side, counterfactual augmentation can expose causal variability missing from purely observational trajectories, helping the policy distinguish causal drivers of reward from spurious predictors (Lu et al., 2020; Chen et al., 2023). Existing approaches often rely on online interaction (Sun et al., 2024a; Cao et al., 2025) or strong structural assumptions such as factored environments with independent entities (Pitis et al., 2022; Urpí et al., 2024), which limits applicability. We instead construct *reward-preserving counterfactual state edits* that do not require environment interaction, do not assume a factored state space, and are designed for standard continuous-control benchmarks such as MuJoCo. On the model side, even with better coverage, a plain Transformer can still allocate capacity to spurious correlations (Hu et al., 2024; Kim et al., 2024). This motivates architectural guidance that reflects the causal roles of the tokens used by DT: the current state and the desired return-to-go should be the primary parents for action prediction, while the remaining context can support longer-horizon reasoning.

Concretely, we propose *CSET* (**C**ounterfactual **S**tate **E**diting **T**ransformer), which combines reward-preserving counterfactual augmentation with a causally guided policy architecture for offline RL. First, CSET learns a *causal reward model* (CRM) that supports abduction by inferring a posterior distribution over latent reward disturbances from offline transitions. Using the factual action and an inferred disturbance sample, a *counterfactual state generator* (CSG) proposes a minimally edited state whose predicted reward matches the factual reward; a move-band constraint and an acceptance gate enforce state plausibility and reward consistency. During augmentation, we replace only the observation token in a trajectory copy; the next observation is kept factual and is *not* treated as the successor of the edited state. This avoids training the policy on synthetic successor transitions while still breaking spurious correlations in the conditioning state. Second, CSET uses a *causally structured hybrid architecture*: separate modality-specific convolutional encoders extract local temporal structure from return-to-go, state, and action streams, and a final attention block is softly supervised so that action prediction attends to its direct causal parents (state and return-to-go). Together, these components aim to produce offline policies that are less sensitive to spurious features and generalize more reliably.

Our main contributions are as follows:

- We introduce CSET, a model-free offline RL method that combines reward-preserving counterfactual state editing with causal architectural guidance.

- We propose a causal reward model and a counterfactual state generator that produce reward-preserving state edits under explicit plausibility and consistency constraints, without requiring environment interaction or a learned dynamics model.

- We develop a causally structured hybrid policy architecture with separate modality-specific encoders and causal supervision for action prediction, encouraging attention to follow the causal roles of DT tokens.

- We evaluate CSET on two distinct sequential decision-making settings (robotic control and recommendation) and show improved robustness and generalization over Transformer-based baselines, including under interventions on spurious distractors.

**Conflict of Interest Disclosure.** The authors declare no financial conflicts of interest related to this work.

## 2. Preliminaries

### 2.1. Decision Transformer (DT)

We consider the offline reinforcement learning (RL) setting, where the agent has access only to a fixed dataset of trajectories and cannot interact with the environment. The environment is modeled as a Markov Decision Process (MDP), defined by $\mathcal{M} = (\mathcal{S}, \mathcal{A}, P, R, \gamma)$, where $\mathcal{S}$ is the state space, $\mathcal{A}$ is the action space, $P(s' \mid s, a)$ specifies the transition dynamics, $R(s, a)$ is the reward function, and $\gamma \in [0, 1)$ is the discount factor.

In offline RL, the dataset $\mathcal{D} = \{\tau_i\}_{i=1}^N$, with $\tau_i = (s_0, a_0, r_0, \ldots, s_T, a_T, r_T)$, is collected under one or more behavior policies, and the agent must learn a new policy without further environment interaction. The Decision Transformer (DT) (Chen et al., 2021) formulates offline RL as conditional sequence modeling. Instead of explicitly estimating value functions or dynamics, DT trains a Transformer autoregressively on trajectory data. Each trajectory is tokenized into a sequence:

$$\tau = (\hat{G}_0, s_0, a_0, \hat{G}_1, s_1, a_1, \ldots, \hat{G}_T, s_T, a_T),$$

where $\hat{G}_t = \sum_{k=t}^T \gamma^{k-t} r_k$ denotes the returns-to-go (RTG). The model is trained to predict the action $a_t$ conditioned on a context window of the most recent $K$ tokens:

$$\pi_\theta(a_t \mid \hat{G}_{t-K+1:t}, s_{t-K+1:t}, a_{t-K+1:t-1}).$$

## 2.2. Structural Causal Model (SCM)

A Structural Causal Model (SCM) provides a formal framework for representing cause-effect relationships (Pearl, 2009). An SCM is a tuple $\mathcal{M}_c = (\mathcal{U}, \mathcal{V}, \mathcal{F}, P(\mathcal{U}))$, where $\mathcal{U}$ is a set of exogenous (external) variables, $\mathcal{V}$ is a set of endogenous (internal) variables, and $\mathcal{F}$ is a set of structural equations. Each equation $V_i := f_i(\text{Pa}(V_i), U_i)$ specifies how an endogenous variable $V_i \in \mathcal{V}$ is determined by its direct causes (parents) $\text{Pa}(V_i) \subseteq \mathcal{V}$ and an exogenous noise variable $U_i \in \mathcal{U}$, drawn from the distribution $P(\mathcal{U})$.

**Causal Relationships in MDPs** The agent-environment interaction loop in an MDP can be formally described by an SCM (Peters et al., 2017; Zhang et al., 2020; Bennett et al., 2021; Shi et al., 2022). The causal mechanisms governing the policy, state transitions, and rewards are given by the following structural equations:

$$
\begin{aligned}
a_t &:= \pi(s_t, u_a), \\
s_{t+1} &:= f_s(s_t, a_t, u_s), \\
r_t &:= f_r(s_t, a_t, u_r),
\end{aligned}
\tag{1}
$$

where $a_t$, $s_{t+1}$, and $r_t$ are endogenous variables. The functions $\pi$, $f_s$, and $f_r$ represent the causal mechanisms for action selection, state transition, and reward generation, respectively. The terms $u_a, u_s,$ and $u_r$ are mutually independent exogenous noise variables that account for the stochasticity in the system.

## 3. The Proposed Method

We propose **CSET** (**C**ounterfactual **S**tate **E**diting **T**ransformer), a framework for improving offline RL by injecting causal consistency at both the data and model levels. The key idea is (i) to enrich an offline dataset with *reward-preserving counterfactual state edits* that break spurious correlations while keeping labels consistent, and (ii) to guide a Transformer policy so that action prediction follows the known causal roles of the tokens. As illustrated in Figure 1, CSET has two components: (i) a *Reward-Preserving Counterfactual State Editing* module that proposes alternative states which, under the factual action and an abduced reward disturbance, would yield the same reward, producing label-preserving tuples $(s_c, a_t, r_t)$; and (ii) a *Causally-Guided Hybrid Policy Architecture* that combines modality-specific local temporal processing with a final global attention block whose action-query attention is softly supervised toward the direct causal parents of the action (the current state and the desired return-to-go). Importantly, CSET is designed for the strictly offline, DT-style setting: it does *not* learn a transition model and does *not* create synthetic successor transitions.

## 3.1. Reward-Preserving Counterfactual State Editing

We use only offline data to answer the counterfactual query: *what other state observation could have produced the same reward under the same action, while holding the unobserved reward disturbance fixed?* We follow Pearl's abduction–action–prediction procedure (Pearl et al., 2000; Pearl, 2009). In strictly offline RL, action-level counterfactuals would require predicting the consequence of new actions, which in turn requires either environment interaction or a reliable learned dynamics model. We therefore intervene on the *state token* and enforce the forms of consistency that are checkable offline: reward consistency under a learned reward SCM and state plausibility via a trust-region constraint.

Concretely, we use a two-stage process: (i) a *Causal Reward Model* (CRM) that supports abduction by inferring a posterior over reward disturbances from offline transitions, and (ii) a *Counterfactual State Generator* (CSG) that proposes a minimally edited state while holding the factual action and the abduced disturbance fixed.

**Causal Reward Model (CRM)** We model the reward mechanism with the SCM equation

$$
r_t = f_r(s_t, a_t, u_r),
$$

where $u_r$ is an unobserved exogenous disturbance. We approximate this mechanism with a conditional variational autoencoder (CVAE), where the latent variable $z$ serves as a *learned proxy* for reward disturbance: rather than recovering the true exogenous noise $u_r$, it parameterizes the family of states that yield the same reward under a given action. Different samples of $z$ thus yield different reward-consistent counterfactuals, which enables diverse label-preserving augmentation downstream. The encoder $q_\phi(z \mid s_t, a_t, r_t)$ performs abduction by learning a posterior distribution over disturbances consistent with the observed transition, and the decoder $p_\theta(r_t \mid s_t, a_t, z)$ represents the learned structural function $\hat{f}_r$. The model is trained by maximizing the ELBO:

$$
\begin{aligned}
\mathcal{L}_{\text{CVAE}} = \ &\mathbb{E}_{q_\phi(z|s_t,a_t,r_t)} \left[ \log p_\theta(r_t \mid s_t, a_t, z) \right] \\
&- \beta \, \text{KL}(q_\phi(z \mid s_t, a_t, r_t) \,\|\, p(z)).
\end{aligned}
\tag{2}
$$

During counterfactual construction, we sample $z_t \sim q_\phi(z \mid s_t, a_t, r_t)$ to keep the abduced disturbance fixed in the counterfactual world.

**Counterfactual State Generator (CSG)** Given a factual tuple $(s_t, a_t, r_t)$, we first sample an abduced disturbance $z_t \sim q_\phi(z \mid s_t, a_t, r_t)$, and then generate an edited state

$$
s_c = g_\psi(s_t, a_t, z_t).
$$

The CSG is trained to preserve the CRM reward under the factual action and fixed $z_t$, while keeping the edit magnitude

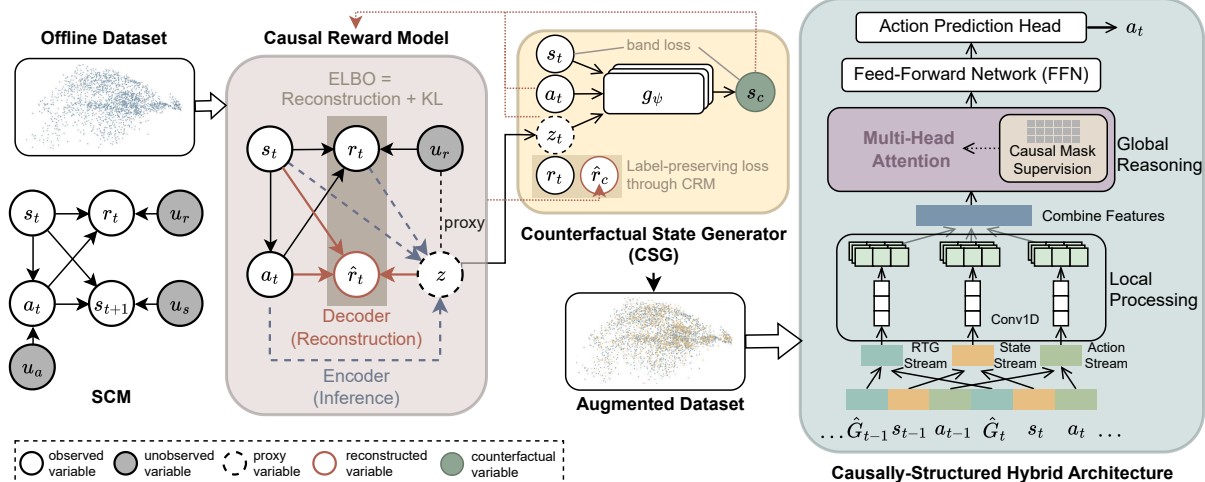

*Figure 1.* Overview of the proposed CSET framework. Left/Middle: A Causal Reward Model (CRM) and Counterfactual State Generator (CSG) jointly produce a causally consistent augmented dataset from offline trajectories. Right: A Causally-Structured Hybrid Architecture processes the data with local convolutions and global, causally supervised attention.

within a controlled band in *normalized* state space. Let $\sigma_s$ be the per-dimension standard deviation of states in the dataset, and define $\Delta_t = (s_c - s_t)/\sigma_s$. We optimize

$$\mathcal{L}_{\text{CSG}} = \left(\hat{f}_r(s_c, a_t, z_t) - r_t\right)^2$$
$$+ \lambda_{\text{band}}\left(\left[\rho_{\text{low}} - \|\Delta_t\|_2\right]_+^2 + \left[\|\Delta_t\|_2 - \rho_{\text{high}}\right]_+^2\right),$$
$$(3)$$

where $[\cdot]_+ = \max(0, \cdot)$. The first term enforces reward preservation under the learned SCM. The move band constrains the normalized edit magnitude $\|\Delta_t\|_2$ to lie within $[\rho_{\text{low}}, \rho_{\text{high}}]$: the lower bound $\rho_{\text{low}}$ prevents trivial near-copies, while the upper bound $\rho_{\text{high}}$ discourages off-manifold edits. Additional details are provided in Appendix G.

**Data Augmentation and Acceptance Gating**  After training CRM and CSG, we generate an augmented dataset by creating a small number of edited copies per trajectory. For selected time steps $t$, we propose $s_c = g_\psi(s_t, a_t, z_t)$ and accept it only if it passes the following gates:

$$\left|\hat{f}_r(s_c, a_t, z_t) - r_t\right| \le \varepsilon_r \quad \text{and} \quad \left\|\frac{s_c - s_t}{\sigma_s}\right\|_2 \le \rho_{\text{high}}.$$
$$(4)$$

If accepted, we replace the observation token $s_t$ with $s_c$ while keeping the action $a_t$ and the reward label $r_t$ unchanged. Crucially, the next observation $s_{t+1}$ is kept factual; we do *not* treat it as the successor of $s_c$. Thus, CSET does not train on synthetic successor transitions $(s_c, a_t, s_{t+1})$. The final dataset is the union of all original trajectories and their edited copies.

**Validity of Counterfactual State Editing**  Sampling $z_t$ from $q_\phi(z \mid s_t, a_t, r_t)$ implements abduction by conditioning on evidence rather than estimating a true exogenous value. Conditioning $g_\psi$ on $(s_t, a_t, z_t)$ ensures edits remain aligned with the action-conditioned reward mechanism, and the move constraint keeps edits near the dataset support.

**Proposition 1** (Reward-consistent accepted edits)**.** *Consider the reward SCM $r = f_r(s, a, u_r)$. Let $(s_t, a_t, r_t)$ be a factual transition with $r_t = f_r(s_t, a_t, u_r)$. Let $z_t$ be the latent inferred from the CRM encoder on $(s_t, a_t, r_t)$. Assume the CRM decoder is locally accurate under the abduced context: for any $s$ such that $\|(s - s_t)/\sigma_s\|_2 \le \rho_{\text{high}}$,*

$$\left|\hat{f}_r(s, a_t, z_t) - f_r(s, a_t, u_r)\right| \le \delta.$$

*If an edited state $s_c$ is accepted by the gates $\left|\hat{f}_r(s_c, a_t, z_t) - r_t\right| \le \varepsilon_r$ and $\|(s_c - s_t)/\sigma_s\|_2 \le \rho_{\text{high}}$, then the factual reward is an approximately valid label for $(s_c, a_t)$:*

$$\left|f_r(s_c, a_t, u_r) - r_t\right| \le \delta + \varepsilon_r.$$

### 3.2. Causally-Guided Hybrid Policy Architecture

Counterfactual state editing improves the training distribution, but the policy model should also be guided away from superficial correlations. We therefore use a hybrid architecture with (i) modality-specific local temporal encoders and (ii) a final global attention block with causal-parent supervision for action prediction. These components are used for policy modeling, not for learning a transition model.

**Local temporal processing with modality-specific encoders** We tokenize trajectories into return-to-go, state, and action tokens. Each modality is processed by its own 1D convolutional encoder. This avoids early fusion of heterogeneous signals and provides a local temporal inductive bias aligned with short-range dependence in trajectory tokens. The convolutional outputs are fused into a shared representation that is passed to the final attention block.

**Global reasoning with causal-parent attention supervision** After convolutional encoding, we apply a final multi-head self-attention block for long-range conditioning. To align action prediction with the known causal structure of DT tokens, we softly supervise the attention for action-token queries toward their direct parents at the same timestep: the current state and the desired return-to-go.

Let $M \in \{0,1\}^{L \times L}$ be a binary mask over the token sequence, where $M[i,j] = 1$ indicates that token $j$ is a valid parent for action-token query $i$. Let $\mathcal{I}_a$ denote the set of indices corresponding to action tokens, and $S_i = \{j : M[i,j] = 1\}$. We define a uniform target distribution $q_{i,j} = \frac{1}{|S_i|}\mathbb{I}[j \in S_i]$, and supervise the attention matrix $A$ (for the supervised heads of the final block) with

$$\mathcal{L}_{\text{mask}} = \frac{1}{|\mathcal{I}_a|} \sum_{i \in \mathcal{I}_a} \sum_{j \in S_i} -q_{i,j} \log(A_{i,j} + \epsilon), \qquad (5)$$

with $\epsilon$ for numerical stability. This is a soft guidance term: it encourages action-token queries to allocate most attention mass to causal parents, while the model remains free to use the full context through other attention patterns and other heads.

Finally, we train the policy with the standard DT action prediction loss $\mathcal{L}_{\text{action}}$ (negative log-likelihood for discrete actions or MSE for continuous actions) plus the mask loss:

$$\mathcal{L}_{\text{total}} = \mathcal{L}_{\text{action}} + \lambda \mathcal{L}_{\text{mask}}.$$

## 4. Experiments

We conduct experiments to evaluate **CSET** and answer five questions: (i) Does CSET outperform strong offline RL baselines on standard D4RL benchmarks? (ii) Does CSET generalize beyond robotic control to sequential recommendation? (iii) Is CSET more robust to spurious correlations under test-time interventions? (iv) What are the contributions of counterfactual state editing and the causal hybrid architecture? (v) Do the generated state edits stay plausible and which safeguards are necessary?

### 4.1. Locomotion and AntMaze Tasks

We first evaluate CSET on a diverse suite of locomotion and navigation tasks from D4RL (RQ i). We evaluate CSET

on locomotion and navigation tasks from D4RL and compare against leading value-based and Decision Transformer baselines.

**Datasets** We use continuous-control datasets from D4RL (Fu et al., 2020). For locomotion, we evaluate on **nine** datasets from three MuJoCo environments (HalfCheetah, Hopper, Walker2d) under three standard regimes: the -medium (m), -medium-replay (m-r), and -medium-expert (m-e) datasets. For navigation, we include Antmaze-umaze (u) and Antmaze-umaze-diverse (u-d). Full details are in Section D.1.

**Baselines** We compare CSET against several state-of-the-art offline RL algorithms, spanning three families: conservative offline RL methods (IQL (Kostrikov et al., 2021) and CQL (Kumar et al., 2020)), value-based methods (ReBRAC (Tarasov et al., 2023) and EDAC (An et al., 2021)), and DT-based methods (the standard DT (Chen et al., 2021), DC (Kim et al., 2024), and LSDT (Wang et al., 2025a)). Full details of the baselines are given in Section D.1.

**Overall Results** The performance of CSET and all baselines is summarized in Table 1, reporting expert-normalized returns following D4RL (Fu et al., 2020). Within the DT family, CSET achieves the highest score on 10 of 11 datasets, with LSDT marginally ahead only on `HalfCheetah-medium-expert` (93.2 vs. 93.1). CSET improves clearly over the standard DT across Hopper, Walker2d, and AntMaze, including the mixed-quality medium-replay and medium-expert regimes, showing effective learning from heterogeneous offline data and from sparse-reward, long-horizon navigation. Across method families, value-based actor-critic baselines (ReBRAC and EDAC) achieve higher absolute scores on several locomotion datasets, but this advantage is not uniform: EDAC fails on AntMaze (both `umaze` and `umaze-d` score 0.0). Since the central claim of CSET's design is robustness to spurious correlations rather than absolute clean-task performance, we test this property directly in Section 4.3.

### 4.2. Recommendation Tasks

**Datasets** For the recommendation domain, we evaluate CSET on three large-scale, real-world datasets: KuaiRand (Gao et al., 2022b), KuaiRec (Gao et al., 2022a), and VirtualTB (Shi et al., 2019). Full details of the datasets are given in Section D.2.

**Baselines** In this domain, we compare CSET against recent DT-based baselines tailored for recommender systems: DT4Rec (Zhao et al., 2023), CDT4Rec (Wang et al., 2023), and EDT4Rec (Chen et al., 2024). Full details of the baselines are given in Section D.2.

*Table 1.* Offline results on the MuJoCo and Antmaze datasets. We report the expert-normalized returns, averaged across 5 random seeds for MuJoCo and 4 for Antmaze. **Bold** denotes the best score overall (within 5% of the best); underline denotes the highest score within the DT family. † denotes value-based actor-critic methods.

| Method | H-Cheetah | | | Hopper | | | Walker2d | | | Antmaze | |
|---|---|---|---|---|---|---|---|---|---|---|---|
| | -m | -m-r | -m-e | -m | -m-r | -m-e | -m | -m-r | -m-e | -umaze | -umaze-d |
| IQL | 47.4 | 44.2 | 86.7 | 66.3 | **94.7** | 91.5 | 78.3 | 73.9 | **109.6** | 87.5 | 62.2 |
| CQL | 44.0 | 37.5 | 91.6 | 58.5 | **95.0** | 105.4 | 72.5 | 77.2 | **108.8** | 74.0 | **84.0** |
| ReBRAC† | **63.0** | 47.1 | **100.5** | 98.4 | 91.3 | 105.2 | 82.1 | **81.4** | 108.9 | 94.1 | 80.5 |
| EDAC† | **64.2** | **58.3** | **100.8** | 96.2 | 92.5 | 100.1 | **89.4** | 83.0 | 109.2 | 0.0 | 0.0 |
| DT | 42.6 | 36.6 | 86.8 | 67.6 | 82.7 | **107.6** | 74.0 | 66.6 | **108.1** | 69.8 | 70.3 |
| DC | 43.0 | 41.3 | 93.0 | 92.6 | **94.2** | 110.4 | 79.2 | 76.6 | **109.6** | 82.2 | 78.5 |
| LSDT | 43.6 | 42.9 | 93.2 | 87.2 | **93.9** | 111.7 | 81.0 | 74.7 | **109.8** | 80.0 | **83.2** |
| **CSET** | 44.9 | 43.5 | 93.1 | 93.4 | **94.8** | **112.0** | 82.5 | 77.0 | **110.7** | 83.1 | **84.5** |

*Table 2.* Evaluation results on recommendation datasets. Metrics include Cumulative Reward ($\mathcal{R}_{cum}$) and Average Reward ($\mathcal{R}_{avg}$). **Bold** indicates the best performance per metric, and * marks the second-best.

| Method | KuaiRand | | KuaiRec | | VirtualTB | |
|---|---|---|---|---|---|---|
| | $\mathcal{R}_{cum}$ | $\mathcal{R}_{avg}$ | $\mathcal{R}_{cum}$ | $\mathcal{R}_{avg}$ | $\mathcal{R}_{cum}$ | $\mathcal{R}_{avg}$ |
| DT4Rec | $6.8172 \pm 2.45$ | $0.5686 \pm 0.21$ | $28.5418 \pm 10.42$ | $0.8798 \pm 0.34$ | $76.7871 \pm 22.63$ | $5.4420 \pm 1.72$ |
| CDT4Rec | $7.3271 \pm 1.98$ | $0.6508^* \pm 0.19$ | $30.4888 \pm 10.19$ | $1.0061 \pm 0.35$ | $79.2101 \pm 22.38$ | $5.6490 \pm 1.61$ |
| EDT4Rec | $7.5817^* \pm 1.84$ | $0.6497 \pm 0.17$ | $31.0726^* \pm 10.98$ | $1.0397^* \pm 0.39$ | $79.6651^* \pm 21.67$ | $5.6741^* \pm 1.54$ |
| CSET | $\mathbf{7.6221} \pm 1.79$ | $\mathbf{0.6554} \pm 0.16$ | $\mathbf{31.8721} \pm 10.55$ | $\mathbf{1.0582} \pm 0.40$ | $\mathbf{80.4241} \pm 21.15$ | $\mathbf{5.6957} \pm 1.51$ |

**Overall Results** The evaluation results on the three recommendation datasets are presented in Table 2. The findings are unequivocal: CSET consistently outperforms all specialized, state-of-the-art Decision Transformer baselines across every metric on all datasets. Specifically, CSET achieves the highest Cumulative Reward ($\mathcal{R}_{cum}$) and Average Reward ($\mathcal{R}_{avg}$) on KuaiRand, KuaiRec, and VirtualTB, often by a clear margin over the second-best methods. This strong performance provides an affirmative answer to our second research question (RQ ii), demonstrating that CSET generalizes effectively beyond robotic control to the distinct domain of sequential recommendation. The ability of our general-purpose model to surpass domain-specialized methods highlights the robustness and broad applicability of our proposed architecture.

### 4.3. Robustness to Spurious Correlations

To test whether CSET mitigates causal confusion (RQ iii), we construct semi-synthetic variants of D4RL benchmarks (HalfCheetah-medium, Hopper-medium, and Walker2d-medium) by appending a binary distractor feature $d_t$ to the state. The distractor is correlated with reward in the offline training data but has no causal effect on the environment. We evaluate two settings. In the *deterministic* setting, $d_t$ is defined from the reward threshold (a perfectly reward-aligned bit in the dataset). In the *stochastic* setting, we flip this bit with a fixed probability to mimic a noisy

*Table 3.* Performance degradation under spurious distractor intervention. Each entry is the percent drop in expert-normalized return when the distractor is fixed to $d_t = 0$ at test time, relative to the clean baseline (no distractor). Lower is better.

| | HalfCheetah-m | | Hopper-m | | Walker2d-m | |
|---|---|---|---|---|---|---|
| Method | Det. | Stoch. | Det. | Stoch. | Det. | Stoch. |
| DT | 20% | 16% | 32% | 30% | 37% | 29% |
| DC | 10% | 5% | 23% | 18% | 15% | 10% |
| ReBRAC | 36% | 23% | 52% | 35% | 44% | 32% |
| EDAC | 21% | 18% | 19% | 15% | 17% | 12% |
| **CSET** | **5%** | **3%** | **6%** | **5%** | **3%** | **2%** |

sensor, which weakens (but does not remove) the correlation. At test time, we intervene by fixing the distractor to $d_t = 0.0$ for every observed state before passing it to the policy, which preserves the input dimension but breaks the learned shortcut.

Figure 2 reports expert-normalized returns (mean ± standard deviation across 5 seeds) under the clean baseline, the deterministic distractor, and the stochastic distractor. To make the cross-method comparison explicit, Table 3 reports the corresponding percentage drop relative to each method's clean baseline. The DT-family baselines DT and DC drop by 16–37% and 5–23% respectively. More strikingly, the value-based baselines ReBRAC and EDAC, which achieved the highest absolute clean-task returns in Table 1, degrade

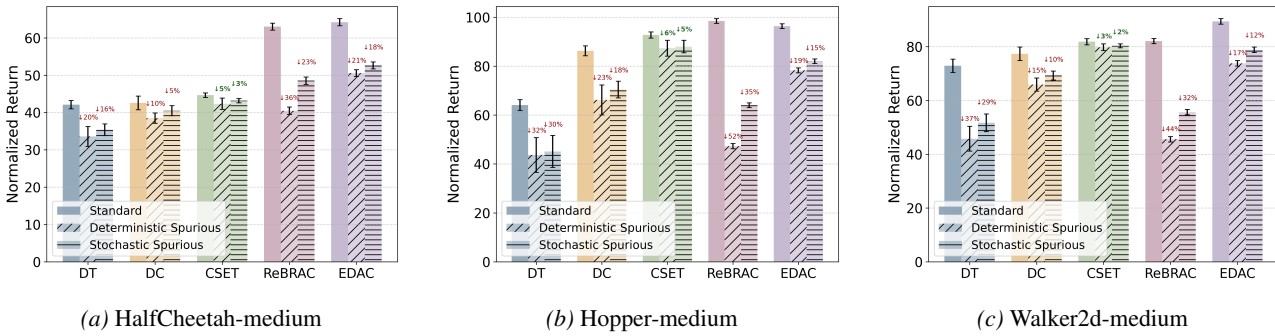

*Figure 2.* Robustness to spurious distractors (deterministic and stochastic). Expert-normalized returns for DT, DC, CSET, ReBRAC, and EDAC. We compare training on the standard dataset (no distractor) to training with a binary distractor that is either deterministic or stochastic. Despite ReBRAC and EDAC achieving the highest clean-task returns, they degrade sharply under the intervention, while CSET stays close to its clean performance.

*Table 4.* Ablation study of CSET components on MuJoCo and Antmaze tasks. We compare the full model against variants without Counterfactual State Editing (w/o Edit) and without the Causally-Structured Hybrid Architecture (w/o Arch). Bold numbers denote the best score and any scores within 5% of the best for each dataset.

| Method | H-Cheetah | | | Hopper | | | Walker2d | | | Antmaze | |
|---|---|---|---|---|---|---|---|---|---|---|---|
| | -m | -m-r | -m-e | -m | -m-r | -m-e | -m | -m-r | -m-e | -umaze | -umaze-d |
| **CSET** | **44.9** | **43.5** | **93.1** | **93.4** | **94.8** | **112.0** | **82.5** | **77.0** | **110.7** | **83.1** | **84.5** |
| w/o Edit | **43.8** | **42.7** | **93.0** | **92.8** | **94.3** | **111.2** | **81.5** | **76.8** | **110.0** | **81.5** | **82.5** |
| w/o Arch | **43.2** | 41.0 | **91.0** | 84.2 | **91.5** | **110.0** | 79.8 | 73.5 | **109.2** | 78.0 | 79.5 |
| DT | 42.6 | 36.6 | 86.8 | 67.6 | 82.7 | **107.6** | 74.0 | 66.6 | **108.1** | 69.8 | 70.3 |
| DC | **43.0** | 41.3 | **93.0** | 92.6 | **94.2** | 110.4 | 79.2 | 76.6 | **109.6** | 82.2 | 78.5 |
| LSDT | **43.6** | **42.9** | **93.2** | 87.2 | **93.9** | **111.7** | 81.0 | 74.7 | **109.8** | 80.0 | 83.2 |

sharply once the spurious feature is removed: ReBRAC drops by 23–52% and EDAC by 12–21% across the three environments and both distractor types. In contrast, CSET remains close to its clean performance under both distractor types, with degradation of only 2–6%, the smallest of any method evaluated.

This pattern supports CSET's central design claim: reward-preserving counterfactual edits combined with causal-parent attention supervision teach the policy to rely on reward-relevant structure rather than spurious shortcuts, even against methods with substantially higher clean-task scores. Full construction details, normalization, and evaluation protocol are provided in Appendix F.4.

### 4.4. Ablation Study

**Contributions of Two Core Components (RQ iv)** To study the role of each component, we compare CSET to two variants: (i) **w/o Edit**, which removes counterfactual state editing, and (ii) **w/o Arch**, which replaces the causal hybrid architecture with a standard DT backbone. The results in Table 4 show complementary effects. On dense-reward MuJoCo tasks, removing the architecture (w/o Arch) leads to a clear drop, especially on the more diverse medium-

replay datasets for Hopper and Walker2d, while removing state editing (w/o Edit) causes smaller but consistent degradations. On sparse-reward AntMaze, w/o Edit hurts more, and w/o Arch drops even further, indicating that both components matter: state editing expands useful training variation, and the causal hybrid architecture helps convert that variation into stable long-horizon policies.

**Effect of Causal Supervision Weight $\lambda$.** We investigate the effect of the causal supervision weight $\lambda$, which balances the action prediction loss and the mask supervision loss. Setting $\lambda = 0$ removes causal supervision entirely, reducing CSET to a standard Transformer; $\lambda = 1$ enforces full adherence to the causal mask. Figure 3 shows that moderate values of $\lambda$ typically yield the best trade-off. In simple environments such as Hopper, larger $\lambda$ improves stability and performance. In more complex domains such as HalfCheetah or Antmaze, too much supervision can constrain the model, and intermediate $\lambda$ values achieve the strongest results. This confirms that causal supervision is beneficial, but its strength should adapt to task complexity.

**Counterfactual State Editing (RQ v).** Beyond ablating the two main components, we analyze the state editing pipeline itself. We study which safeguards are necessary

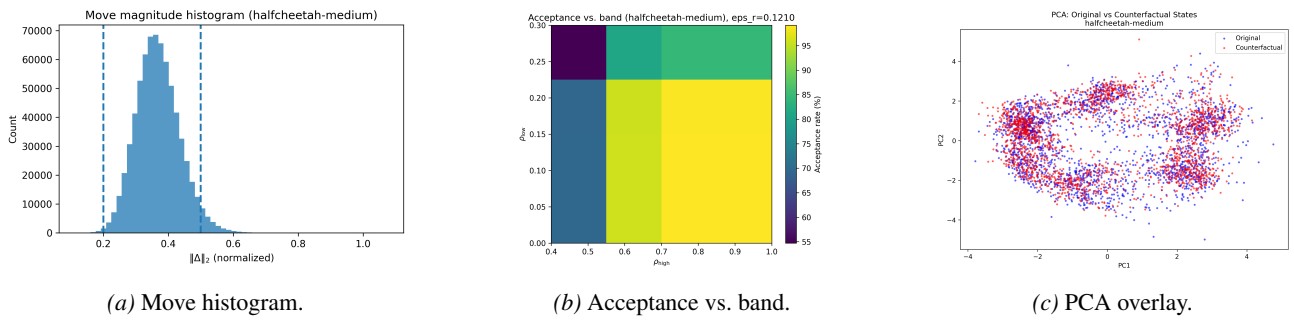

*Figure 3.* Ablation on the causal supervision weight $\lambda$ across MuJoCo locomotion tasks and Antmaze. Each plot reports normalized returns (mean $\pm$ standard deviation over 5 seeds).

*(a)* Move histogram.  *(b)* Acceptance vs. band.  *(c)* PCA overlay.

*Figure 4.* The default band $(0.2, 0.5)$ yields moderate edit sizes, structured acceptance behavior across band settings, and counterfactual states that overlap the dataset support.

(reward-consistency gate and move-band constraint), and we provide diagnostics to verify that accepted edits are non-trivial yet remain close to the dataset support. Figure 4 visualizes the edit magnitudes, acceptance rates under band sweeps, and a PCA overlay of original versus edited states. Full results are reported in Appendix H.

## 5. Related Work

**Transformer Architectures for Reinforcement Learning.** Recent work has reframed sequential decision making as a sequence modeling problem, where trajectories of returns-to-go, states, and actions are treated as tokens processed by Transformer-based architectures. DT (Chen et al., 2021) demonstrated that conditioning on return-to-go enables policy learning without value functions or explicit dynamics models. Trajectory Transformer (Janner et al., 2021) extended this perspective by discretizing trajectories and using beam search for planning. Beyond purely autoregressive modeling, masked prediction has been explored as an alternative training paradigm. Uni[MASK] (Carroll et al., 2022) proposed a unified framework where diverse inference tasks are cast as different masking patterns, while MaskDP (Liu et al., 2022) applied masked autoencoding to state–action trajectories, showing strong zero-shot and fine-tuning performance. At the architectural level, several works have explored combining attention with convolution to better align with the structure of decision-making problems. DC (Kim et al., 2024) replaced most attention blocks with lightweight

convolutional mixers, showing that local filtering is often sufficient for Markovian dynamics while retaining a final attention block for long-range reasoning. LSDT (Wang et al., 2025a) proposed a parallel hybrid design where each block splits into a convolution branch for local dependencies and an attention branch for global dependencies, with channel ratios controlling their relative contributions. This sequence modeling paradigm has also been extended to recommendation tasks. CDT4Rec (Wang et al., 2023) incorporated a causal reward estimator, EDT4Rec (Chen et al., 2024) leveraged entropy regularization and reward relabeling for learning from suboptimal data, and MaskRDT (Wang et al., 2025b) improved efficiency on long user histories through retentive networks and adaptive masking. Our approach CSET, differs from prior architectures by its explicit causal grounding. Unlike DC and LSDT, which combine convolution and attention without causal guidance, CSET augments offline data with counterfactual trajectories that preserve causal consistency and constrains attention to focus on true causal parents. This integration of causal augmentation and causal priors yields policies that are more robust and generalizable.

**Causal Reinforcement Learning.** A complementary line of research introduces causality into RL to address spurious correlations and improve generalization (Zeng et al., 2025). CDL (Wang et al., 2022) learns task-independent state abstractions by uncovering the causal structure of environment dynamics. ACE (Ji et al., 2024) proposes causality-aware entropy regularization that weights action

dimensions by their causal influence on rewards, improving exploration efficiency in continuous control. CSR (Yang et al., 2025) develops causality-guided self-adaptive representations that detect distribution shifts, expand causal graphs to accommodate new variables, and prune irrelevant factors, enabling more generalizable policy transfer. A particularly active direction focuses on counterfactual reasoning. MOCODA (Pitis et al., 2022) generates counterfactual transitions using a factored dynamics model for improved out-of-distribution generalization. CAIAC (Urpí et al., 2024) targets robotic manipulation, augmenting data by swapping action-independent factors between trajectories. ACAMDA (Sun et al., 2024b) applies adversarial counterfactual augmentation to enforce causally consistent dynamics. More recently, CIP (Cao et al., 2025) combines counterfactual data augmentation with causality-aware empowerment to improve sample efficiency across domains. Unlike prior counterfactual augmentation methods, which either assume factored environments with independent entities (MOCODA, CAIAC) or rely on online interaction with the environment (ACAMDA, CIP), our method is explicitly tailored for strictly offline RL where only a fixed dataset is available. Crucially, we do not assume a factorized state space, making our approach applicable to standard continuous-control benchmarks such as Mujoco. And we further introduce a causally structured hybrid architecture with supervised attention, ensuring that the agent can better benefit from counterfactual augmentation.

## 6. Conclusion

We presented **CSET** (Counterfactual State Editing Transformer), a framework for offline reinforcement learning that improves causal reliability at both the data and model levels. On the data side, CSET learns a *causal reward model* to infer a posterior over latent disturbance factors from offline transitions, and uses a *counterfactual state editor* to propose minimal state edits that preserve the factual reward under the same action and inferred disturbance. Accepted edits pass reward-consistency and trust-region checks, which lets us expand the training distribution without any online interaction and without assuming a factored environment. On the model side, CSET uses a causally guided hybrid policy architecture: modality-specific convolutional encoders capture local structure, while a final attention layer is softly supervised so that action prediction emphasizes its direct causal parents (current state and return-to-go). Across robotic control, navigation, and sequential recommendation, CSET improves generalization and robustness, and it is substantially more stable under both deterministic and stochastic spurious distractors than correlation-driven sequence models.

**Limitations and Future Work.** First, counterfactual state edits depend on the learned reward model, and the local-accuracy assumption is checkable only within the trust region; uncertainty-aware filtering (e.g., ensemble disagreement or posterior variance from a Bayesian reward model) could replace the fixed tolerance $\varepsilon_r$ with a per-proposal confidence test. Second, CSET introduces a small number of tunable scalars (edit band and attention supervision weight) that currently require modest per-task tuning; automatic schedules and extension to higher-dimensional or partially observable settings are natural next steps.

## Impact Statement

This paper introduces CSET, a method for strictly offline reinforcement learning that improves reliability under distribution shift. CSET targets a common failure mode in offline datasets: models can latch onto features that correlate with reward in logged data but are not causally relevant at deployment. By combining reward-preserving counterfactual state edits with causal guidance in the policy architecture, CSET aims to reduce this shortcut behavior and improve generalization when spurious correlations change at test time.

CSET can reduce the need for online trial-and-error because it is trained purely from logged data. This is useful when exploration is costly, slow, or unsafe, such as robotics and other real-world control systems. In sequential recommendation, improving robustness to spurious correlations can also help models rely less on unstable signals and behave more consistently under changes in logging policies or user populations. More broadly, the paper shows one way to incorporate causal structure into Transformer-based decision making without requiring online interventions or full causal discovery.

CSET does not remove the need for careful dataset and reward design. If the logged data encode harmful biases, or if the reward proxy is misaligned with human goals (for example, over-optimizing engagement in recommendation), then improving robustness may still reinforce undesirable behavior. Like other offline RL approaches, CSET can be misapplied in high-stakes settings without adequate validation, monitoring, and safeguards.

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

# A. Background on Causal Inference

Here, we introduce fundamental causal modeling concepts (Pearl, 2009; Peters et al., 2017) that underpin our methodology and theoretical analysis.

**Structural Causal Models**   Structural Causal Models (SCMs) formalize the data-generating process by specifying how each variable is causally determined by its parents and exogenous noise. An SCM is associated with a directed acyclic graph (DAG) that encodes causal relationships among variables.

**Definition A.1** (Structural Causal Model (Pearl, 2009)). A Structural Causal Model (SCM) $\mathcal{M} = (\mathcal{G}, \mathbf{S}, P_\mathbf{U})$ consists of:

- A directed acyclic graph (DAG) $\mathcal{G} = (\mathbf{V}, \mathcal{E})$, where $\mathbf{V}$ is a set of endogenous variables and $\mathcal{E}$ is the set of directed edges representing direct causal relationships;

- A collection of structural assignments $\mathbf{S} = \{X_i = f_i(\mathrm{PA}_i, U_i)\}$ for each $X_i \in \mathbf{V}$, where $\mathrm{PA}_i \subseteq \mathbf{V} \setminus \{X_i\}$ are the parent variables of $X_i$ in $\mathcal{G}$, and $U_i \in \mathbf{U}$ are exogenous noise variables;

- A joint distribution $P_\mathbf{U}$ over the exogenous variables $\mathbf{U} = \{U_1, \ldots, U_n\}$, typically assumed to be mutually independent.

An SCM $\mathcal{M}$ induces a joint observational distribution over $\mathbf{V}$ according to the structural assignments and exogenous distribution.

**Definition A.2** (Intervention (Pearl, 2009)). An *intervention* in an SCM $\mathcal{M} = (\mathcal{G}, \mathbf{S}, P_\mathbf{U})$ corresponds to replacing the structural assignment for a variable $X_j \in \mathbf{V}$ with a new mechanism:

$$X_j = \hat{f}_j(\widehat{\mathrm{PA}}_j, \hat{U}_j),$$

resulting in a modified model $\hat{\mathcal{M}}$. The new model $\hat{\mathcal{M}}$ induces a different distribution over the variables $\mathbf{V}$, referred to as the *interventional distribution*:

$$P_{\hat{\mathcal{M}}}(\mathbf{V}) = P_\mathcal{M}(\mathbf{V} \mid \mathrm{do}(X_j = \hat{f}_j(\widehat{\mathrm{PA}}_j, \hat{U}_j))).$$

**Definition A.3** (Causal Effect Identifiability (Pearl, 2009)). The causal effect of $X$ on $Y$ is *identifiable* from a graph $\mathcal{G}$ if the quantity $P(y \mid \mathrm{do}(x))$ can be computed uniquely from any positive probability of the observed variables. That is, if

$$P_{M_1}(y \mid \mathrm{do}(x)) = P_{M_2}(y \mid \mathrm{do}(x))$$

for every pair of models $M_1$ and $M_2$ such that $P_{M_1}(\nu) = P_{M_2}(\nu) > 0$ and $\mathcal{G}(M_1) = \mathcal{G}(M_2) = \mathcal{G}$.

**Theorem A.4** (Three Steps of Counterfactual Reasoning (Pearl, 2009)). *Computing a counterfactual requires a three-step process:*

1. **Abduction:** *Condition the distribution of the exogenous variables $u_r$ on the observed evidence $e$, obtaining $P(u_r \mid e)$.*

2. **Action:** *Modify the SCM by performing a surgical intervention using the do()-operator, forcing variables to take hypothetical values (e.g., setting the state to $s_c$).*

3. **Prediction:** *Use the modified model together with $P(u_r \mid e)$ to compute the counterfactual outcome.*

# B. Assumptions and Propositions

## B.1. Assumptions

**A1 (Markov property) (Pearl, 2009).** The environment is Markovian. The joint over $\mathcal{V} = \{s_t, a_t, r_t, s_{t+1}\}$ factorizes as $p(s_t)\, p(a_t \mid s_t)\, p(r_t \mid s_t, a_t)\, p(s_{t+1} \mid s_t, a_t)$. *Use:* underlies the SCM in Eq. (1), CRM training, and the counterfactual edit that changes only $s_t$.

**A2 (Faithfulness) (Pearl, 2009).** The observed distribution is faithful to the causal DAG $\mathcal{G}$: conditional independences coincide with $d$-separations (no cancellation). *Use:* justifies that the supervised attention mask encodes true parent sets of $a_t$.

**A3 (Causal minimality)** (Pearl, 2009). The DAG has no redundant edges: for every edge $X \to Y$, $X \not\perp\!\!\!\perp Y \mid \mathrm{Pa}(Y) \setminus \{X\}$. *Use:* prevents spurious parents in the mask and supports interpretability.

**A4 (Temporal causality).** No backward or instantaneous cycles; edges respect time order. *Use:* supports abduction at time $t$ and interventions on $s_t$ while holding $a_t$ and exogenous variables fixed.

### B.2. Proposition on Causal Consistency

We restate Proposition 1 in an assumption–guarantee form. The goal of this proposition is deliberately *local*: in strictly offline RL, we can check and enforce reward consistency for one-step edits, but we cannot verify full transition validity without online interaction or a reliable dynamics model. The argument follows Pearl's abduction–action–prediction view of counterfactuals.

**Setup.** Consider the reward structural equation (SCM)

$$r = f_r(s, a, u_r),$$

where $u_r$ denotes exogenous factors for the reward mechanism. CSET learns a reward predictor $\hat{f}_r(s, a, z)$, where $z$ is a latent variable inferred from the CRM encoder (serving as a proxy for the abduced exogenous context). For a factual tuple $(s_t, a_t, r_t)$, CSET samples $z_t \sim q_\phi(z \mid s_t, a_t, r_t)$ and proposes an edited state $s_c$.

**Additional conditions.**

1. **Local CRM accuracy under the abduced context.** Let $(s_t, a_t, r_t)$ be a factual sample with $r_t = f_r(s_t, a_t, u_r)$, and let $z_t \sim q_\phi(z \mid s_t, a_t, r_t)$ be the sampled latent from the CRM posterior. Assume that for any $s$ within the trust region around $s_t$,

$$\left\| \frac{s - s_t}{\sigma_s} \right\|_2 \leq \rho_{\text{high}} \quad \implies \quad \left| \hat{f}_r(s, a_t, z_t) - f_r(s, a_t, u_r) \right| \leq \delta.$$

   This condition is local-in-support: it only requires the CRM to be accurate in the neighborhood where we allow edits.

2. **Generator proposal and move bound.** The Counterfactual State Generator (CSG) proposes $s_c = g_\psi(s_t, a_t, z_t)$, and candidates are restricted to satisfy the same upper move bound used by the acceptance test.

3. **Acceptance test.** A proposed edited state $s_c$ is accepted only if it satisfies

$$\left| \hat{f}_r(s_c, a_t, z_t) - r_t \right| \leq \varepsilon_r \quad \text{and} \quad \left\| \frac{s_c - s_t}{\sigma_s} \right\|_2 \leq \rho_{\text{high}}.$$

**Guarantee (reward-consistent accepted edits).** Under the above conditions, every accepted edited tuple $(s_c, a_t, r_t)$ is reward-consistent with the reward SCM under the intervention $do(s_t \leftarrow s_c)$ up to error $\delta + \varepsilon_r$. More precisely, define the (SCM) counterfactual reward under the fixed action and fixed exogenous context as

$$r_t^{\text{cf}} \triangleq f_r(s_c, a_t, u_r).$$

Then every accepted $s_c$ satisfies

$$\left| r_t^{\text{cf}} - r_t \right| \leq \delta + \varepsilon_r.$$

Equivalently, $(s_c, a_t, r_t)$ satisfies the reward structural equation under $do(s_t \leftarrow s_c)$ within tolerance $\delta + \varepsilon_r$, which is the form of causal consistency that is directly checkable from offline data.

**Argument.** *Abduction.* Given $(s_t, a_t, r_t)$, the CRM encoder yields a posterior $q_\phi(z \mid s_t, a_t, r_t)$; sampling $z_t$ serves as a proxy for abducing the reward-relevant exogenous context.

*Action.* The CSG produces $s_c = g_\psi(s_t, a_t, z_t)$, corresponding to the intervention $do(s_t \leftarrow s_c)$ while keeping $a_t$ and $u_r$ fixed.

*Prediction.* By the local accuracy condition and the move bound in the acceptance test (which ensures $s_c$ lies in the trust region),

$$\left| \hat{f}_r(s_c, a_t, z_t) - f_r(s_c, a_t, u_r) \right| \le \delta.$$

*Acceptance and conclusion.* Acceptance additionally enforces $\left| \hat{f}_r(s_c, a_t, z_t) - r_t \right| \le \varepsilon_r$. Combining the two inequalities with the triangle inequality gives

$$\left| f_r(s_c, a_t, u_r) - r_t \right| \ \le \ \left| f_r(s_c, a_t, u_r) - \hat{f}_r(s_c, a_t, z_t) \right| + \left| \hat{f}_r(s_c, a_t, z_t) - r_t \right| \ \le \ \delta + \varepsilon_r,$$

which proves the stated guarantee.

**Remark (what this does *not* claim).** This proposition is about one-step reward consistency under a state intervention. CSET does not claim that $(s_c, a_t, s_{t+1})$ is a valid dynamics transition; during augmentation we replace only the observation token and keep $s_{t+1}$ factual (not treated as the successor of $s_c$).

## C. Extended Related Work

### C.1. Causality in Attention Mechanisms

Recent work has explored integrating causality into attention to enhance interpretability and generalization. Some studies interpret attention through a causal lens, such as (Rohekar et al., 2023), who treat self-attention as estimating an SCM via constraint-based methods. Others embed attention into causal frameworks, like CAL (Sui et al., 2022) and CAL+ (Sui et al., 2024), which use attention to identify causal features in GNNs. Intervention-based designs have also emerged, e.g., (Ge et al., 2023) introduce Social Cross Attention with learnable variables representing confounder strata to deconfound human trajectory prediction. In contrast, our approach directly supervises attention heads using a predefined causal graph, aligning attention patterns with known structural dependencies in sequential decision-making tasks.

## D. Datasets and Baselines

### D.1. Robotic Control Task

**Datasets** For robotic control experiments, we use datasets from the D4RL benchmark (Fu et al., 2020), which are widely used in offline reinforcement learning research. These datasets are generated using the MuJoCo physics simulator and consist of pre-collected trajectories from various environments and policy qualities. Specifically, we evaluate on:

- `HalfCheetah`: A 2D bipedal cheetah-like robot aiming to run.

- `Hopper`: A 2D one-legged hopper robot aiming to hop forward.

- `Walker2d`: A 2D bipedal robot aiming to walk.

For each of these environments, we use three dataset types reflecting different data quality and collection strategies:

- `-medium (m)`: Trajectories collected by a policy trained to a medium level of performance and then rolled out.

- `-medium-replay (m-r)`: The full replay buffer contents of an agent trained to a medium level of performance.

- `-medium-expert (m-e)`: A 50/50 mix of trajectories from a medium policy and an expert policy.

**Baselines** We compare CSET against several state-of-the-art and representative offline RL algorithms in the robotic control domain.

- **Conservative Offline RL Methods:**
    - `IQL (Implicit Q-Learning)` (Kostrikov et al., 2021): An offline Q-learning method that learns Q-functions by implicitly defining them via expectile regression, avoiding explicit policy constraints or out-of-distribution action queries.

- CQL (Conservative Q-Learning) (Kumar et al., 2020): A widely used offline RL algorithm that learns a conservative Q-function by adding a regularization term to the standard Bellman error. This term penalizes high Q-values for actions outside the dataset distribution and encourages low Q-values for them, mitigating overestimation issues.

- **Recent Value-Based Methods:**

  - ReBRAC (Revisited Behavior-Regularized Actor-Critic) (Tarasov et al., 2023): A minimalist actor-critic approach that revisits behavior-regularized offline RL with carefully tuned design choices (deeper networks, layer normalization, decoupled penalties, and large discount factors), achieving strong performance on D4RL locomotion without ensembles or explicit conservatism.

  - EDAC (Ensemble-Diversified Actor-Critic) (An et al., 2021): An uncertainty-based offline RL method that uses a diversified Q-ensemble to penalize out-of-distribution actions through ensemble disagreement. EDAC reduces the number of Q-networks needed for reliable uncertainty estimation while maintaining strong locomotion performance.

- **Decision Transformer (DT) and Variants:**

  - DT (Decision Transformer) (Chen et al., 2021): The standard Decision Transformer that models RL as a sequence modeling problem, predicting actions autoregressively based on desired returns-to-go, past states, and actions.

  - DC (Decision ConvFormer) (Kim et al., 2024): A DT variant that integrates convolutional layers to better capture local features and short-range temporal dependencies in trajectory tokens, while retaining a final attention block for long-range reasoning.

  - LSDT (Long-Short Decision Transformer) (Wang et al., 2025a): A Decision Transformer variant that augments the model with a long–short temporal module, enabling it to capture both short-term transitions and long-range dependencies more effectively.

## D.2. Recommendation Task

**Datasets**   For evaluating CSET on recommendation tasks, we selected the following publicly available datasets, known for their scale and real-world relevance:

- KuaiRand: An unbiased sequential recommendation dataset collected from the recommendation logs of the Kuaishou video-sharing mobile app (Gao et al., 2022b). It is notable for including millions of intervened interactions where items were randomly exposed within standard recommendation feeds, which helps in studying and mitigating exposure bias. It provides rich side information, including user IDs, interaction timestamps, and features for users and items, across various collection policies.

- KuaiRec: Another dataset from Kuaishou, KuaiRec is distinguished by its "fully-observed" user-item interaction matrix for a subset of users and items, meaning nearly all preferences are known (Gao et al., 2022a). This dense interaction data (e.g., 1,411 users and 3,327 items with 99.6% density in its "small matrix") is valuable for evaluating recommendation models without suffering severely from missing data issues, and for research in unbiased recommendation, interactive RL, and off-policy evaluation. It also contains a larger, sparser "big matrix" and side information like item categories and a social network.

- VirtualTB (Virtual Taobao): An online simulation platform that mimics a real-world e-commerce environment (Taobao) for developing and testing recommender systems (Shi et al., 2019). It is trained on hundreds of millions of real user data points and generates virtual customers with dynamic and static features. VirtualTB allows RL agents to interact with the simulated environment, receive feedback (e.g., clicks), and be evaluated on metrics like Click-Through Rate (CTR).

**Baselines**   For the recommendation tasks, we compare CSET with the following state-of-the-art Decision Transformer-based models designed for recommender systems:

- `CDT4Rec (Causal Decision Transformer for Recommender Systems)` (Wang et al., 2023): This model adapts the Decision Transformer framework for recommendation by incorporating a causal mechanism. It aims to address the challenge of reward function design by estimating rewards based on the causal relationships inferred from user behavior within the transformer architecture.

- `DT4Rec (Decision Transformer for Recommender Systems)` (Zhao et al., 2023): This approach applies the Decision Transformer to focus on user retention in recommender systems . It often employs specific reward prompting strategies tailored for recommendation scenarios to guide the DT model.

- `EDT4Rec (Max-Entropy enhanced Decision Transformer with Reward Relabeling for Offline RLRS)` (Chen et al., 2024): This model enhances Decision Transformer-based methods for recommendation by tackling limitations such as "stitching" suboptimal trajectories and insufficient online exploration. It integrates max-entropy regularization to encourage exploration and a reward relabeling technique (often based on learned Q-values from methods like CQL) to improve learning from suboptimal data.

## E. Algorithms for CSET

---

**Algorithm 1** CSET: End-to-end training on augmented data

---

**Require:** Offline dataset $\mathcal{D}$; CRM encoder $q_\phi$ and decoder $p_\theta$; CSG $g_\psi$; CSET policy model $\pi_\omega$ (hybrid conv + final attention); mask weight $\lambda$; augmentation budget $K$ per trajectory; thresholds $(\varepsilon_r, \rho_{\text{high}})$; state normalization $\sigma_s$; optimizers $\text{Opt}_\phi, \text{Opt}_\theta, \text{Opt}_\psi, \text{Opt}_\omega$

1: **Train CRM** $(q_\phi, p_\theta)$ on $\mathcal{D}$ by maximizing the ELBO (Alg. 2)
2: **Train CSG** $g_\psi$ with banded move and reward-preservation (Alg. 3)
3: **Augment** $\mathcal{D}$ using $(q_\phi, p_\theta, g_\psi)$ and the acceptance gate (Alg. 5, 4); get $\mathcal{D}_{\text{aug}}$
4: **Train CSET policy** $\pi_\omega$ on $\mathcal{D}_{\text{aug}}$ with action loss + mask loss (Alg. 6)
5: **return** trained policy $\pi_\omega$

---

**Algorithm 2** Training the Causal Reward Model (CRM) as a CVAE

---

**Require:** Offline dataset $\mathcal{D} = \{(s_t, a_t, r_t)\}$; prior $p(z)$ (e.g., $\mathcal{N}(0, I)$); encoder $q_\phi(z \mid s, a, r)$; decoder $p_\theta(r \mid s, a, z)$; ELBO weight $\beta$; optimizer $\text{Opt}$

1: Initialize parameters $\phi, \theta$
2: **while** not converged **do**
3:     Sample minibatch $\{(s_t, a_t, r_t)\}_{t=1}^B \sim \mathcal{D}$
4:     Encode: $(\mu_\phi, \Sigma_\phi) \leftarrow q_\phi(z \mid s_t, a_t, r_t)$
5:     Sample latent $z_t \sim \mathcal{N}(\mu_\phi, \Sigma_\phi)$ {via reparameterization}
6:     Decode: $\hat{r}_t \sim p_\theta(r \mid s_t, a_t, z_t)$
7:     ELBO objective:

$$\mathcal{L}_{\text{CVAE}} = \frac{1}{B} \sum_{t=1}^B \Big( \log p_\theta(r_t \mid s_t, a_t, z_t) - \beta\, \text{KL}\big(q_\phi(z \mid s_t, a_t, r_t) \,\|\, p(z)\big) \Big)$$

8:     Update $\phi, \theta \leftarrow \phi, \theta + \eta \nabla_{\phi,\theta} \mathcal{L}_{\text{CVAE}}$ with $\text{Opt}$
9: **end while**
10: **return** trained CRM: encoder $q_\phi$, decoder $p_\theta$ (defining $\hat{f}_r(s, a, z)$)

---

---

**Algorithm 3** Training the Counterfactual State Generator (CSG)

---

**Require:** Trained CRM $(q_\phi, p_\theta)$; generator $g_\psi$; state normalization $\sigma_s$; band radii $(\rho_{\text{low}}, \rho_{\text{high}})$; weights $(\lambda_r, \lambda_{\text{band}})$; optimizer Opt
1: Initialize parameters $\psi$
2: **while** not converged **do**
3:     Sample minibatch $\{(s_t, a_t, r_t)\}_{t=1}^{B}$
4:     Abduction: $z_t \leftarrow \mu_\phi(s_t, a_t, r_t)$ {posterior mean from CRM encoder}
5:     Generate counterfactual proposal: $s_c \leftarrow g_\psi(s_t, a_t, z_t)$
6:     Normalized move: $\Delta_t \leftarrow (s_c - s_t)/\sigma_s$
7:     Reward consistency:
$$L_r = \frac{1}{B} \sum_t \left( \hat{f}_r(s_c, a_t, z_t) - r_t \right)^2$$

8:     Band penalty:
$$L_{\text{band}} = \frac{1}{B} \sum_t \left( [\rho_{\text{low}} - \|\Delta_t\|_2]_+^2 + [\|\Delta_t\|_2 - \rho_{\text{high}}]_+^2 \right)$$

9:     Total loss: $\mathcal{L}_{\text{CSG}} = \lambda_r L_r + \lambda_{\text{band}} L_{\text{band}}$
10:    Update $\psi \leftarrow \psi - \eta \nabla_\psi \mathcal{L}_{\text{CSG}}$ with Opt
11: **end while**
12: **return** trained generator $g_\psi$

---

**Algorithm 4** Acceptance gate for a counterfactual proposal

---

**Require:** $(s_t, a_t, r_t)$; posterior mean $z_t = \mu_\phi(s_t, a_t, r_t)$; proposal $s_c = g_\psi(s_t, a_t, z_t)$; thresholds $(\varepsilon_r, \rho_{\text{high}})$; normalization $\sigma_s$
1: Reward proximity: $\Delta r \leftarrow \left| \hat{f}_r(s_c, a_t, z_t) - r_t \right|$
2: Move size: $m \leftarrow \left\| \frac{s_c - s_t}{\sigma_s} \right\|_2$
3: **if** $\Delta r \leq \varepsilon_r$ **and** $m \leq \rho_{\text{high}}$ **then**
4:     **return** ACCEPT
5: **else**
6:     **return** REJECT
7: **end if**

---

---

**Algorithm 5** Offline counterfactual augmentation

---

**Require:** Dataset $\mathcal{D} = \{\tau\}$; trained CRM $(q_\phi, p_\theta)$; trained CSG $g_\psi$; budget $K$; thresholds $(\varepsilon_r, \rho_{\text{high}})$; normalization $\sigma_s$

1:  $\mathcal{D}_{\text{aug}} \leftarrow \mathcal{D}$ {Start from the original dataset}
2: **for all** trajectory $\tau \in \mathcal{D}$ **do**
3:     Initialize counter $c \leftarrow 0$
4:     **for all** time indices $t$ in $\tau$ **in random order do**
5:       **if** $c \geq K$ **then**
6:         **break**
7:       **end if**
8:       Compute $z_t \leftarrow \mu_\phi(s_t, a_t, r_t)$
9:       Propose $s_c \leftarrow g_\psi(s_t, a_t, z_t)$
10:      **if** Accept$(s_t, a_t, r_t, z_t, s_c)$ **then**
11:        Replace $s_t$ by $s_c$ in a copy of $\tau$ to form $\tilde{\tau}$
12:        Append modified trajectory $\tilde{\tau}$ to $\mathcal{D}_{\text{aug}}$
13:        $c \leftarrow c + 1$
14:      **end if**
15:     **end for**
16: **end for**
17: **return** $\mathcal{D}_{\text{aug}}$

---

---

**Algorithm 6** Train CSET policy on augmented dataset

---

**Require:** $\mathcal{D}_{\text{aug}}$; hybrid model $\pi_\omega$ with modality-specific conv blocks and a final multi-head attention layer; mask weight $\lambda$; optimizer $\text{Opt}_\omega$

1: **while** not converged **do**
2:     Sample trajectory windows (RTG, states, actions) from $\mathcal{D}_{\text{aug}}$
3:     Encode with modality-specific 1D convolutional blocks (RTG/state/action)
4:     Apply final multi-head self-attention to the token sequence
5:     **Action loss** $\mathcal{L}_{\text{action}}$: negative log-likelihood (discrete) or MSE (continuous) for $a_t$
6:     **Mask loss** $\mathcal{L}_{\text{mask}}$: cross-entropy to target $q_{i,j} \propto \mathbf{1}[j \in S_i]$ where $S_i$ are causal parents (state and RTG at time $t$)
7:     Total loss: $\mathcal{L}_{\text{total}} \leftarrow \mathcal{L}_{\text{action}} + \lambda \mathcal{L}_{\text{mask}}$
8:     Descend $\nabla_\omega \mathcal{L}_{\text{total}}$ with $\text{Opt}_\omega$
9: **end while**
10: **return** $\pi_\omega$

---

## F. Implementation Details

We implement CSET on top of the official Decision Transformer codebase[1], incorporating (i) a *Causal Reward Model* (CRM) trained as a conditional VAE, (ii) a *Counterfactual State Generator* (CSG) trained to edit states under causal constraints, and (iii) a *hybrid causal policy architecture* that combines modality-specific convolutions for local dynamics with a final self-attention layer trained with a causal mask loss. Unless otherwise stated, all experiments follow the D4RL protocol with expert-normalized returns.

### F.1. Causal Reward Model (CRM)

The CRM is parameterized as a conditional variational autoencoder. The encoder $q_\phi(z \mid s, a, r)$ and decoder $p_\theta(r \mid s, a, z)$ are implemented as two-layer MLPs with hidden size 256 and ReLU activations. The latent variable dimension is 16 for MuJoCo and 32 for AntMaze. We optimize the ELBO with $\beta = 0.1$, which balances reward reconstruction accuracy and latent regularization. Larger values (e.g., $\beta = 1.0$) led to worse reward prediction in preliminary runs. We use Adam with a learning rate of $3 \times 10^{-4}$, weight decay $10^{-4}$, and batch size 256. Training runs for $10^6$ steps with early stopping.

---

[1] https://github.com/kzl/decision-transformer

*Table 5.* Hyperparameters for the Causal Reward Model (CRM).

| Hyperparameter | Value |
|---|---|
| Latent dimension | 16 (MuJoCo) / 32 (Antmaze) |
| Network | 2-layer MLP (ReLU, 256) |
| Optimizer | Adam |
| Learning rate | $3 \times 10^{-4}$ |
| Weight decay | $1 \times 10^{-4}$ |
| Batch size | 256 |
| KL weight $\beta$ | 0.1 |
| Training epochs | 40 |

## F.2. Counterfactual State Generator (CSG)

The generator $g_\psi(s, a, z)$ is a three-layer MLP (hidden size 256, ReLU). States are normalized by dataset statistics and the move band acts in normalized space. The loss combines reward consistency under fixed $(a, z)$ and a band penalty that encourages $\|\Delta_t\|_2 = \|(s_c - s_t)/\sigma_s\|_2$ to lie in $[\rho_{\text{low}}, \rho_{\text{high}}]$. We train with Adam (learning rate $5 \times 10^{-4}$, batch size 256) for 40 epochs. At acceptance, a counterfactual is kept if $|\hat{r}(s_c, a, z) - r| \leq \varepsilon_r$ (adaptive: $0.1 \times$ reward std if unspecified) and $\|\Delta_t\|_2 \leq \rho_{\text{high}}$.

*Table 6.* Hyperparameters for the Counterfactual State Generator (CSG).

| Hyperparameter | Value |
|---|---|
| Network | 3-layer MLP (ReLU, 256) |
| Optimizer | Adam |
| Learning rate | $5 \times 10^{-4}$ |
| Batch size | 256 |
| Training epochs | 40 |
| Move-band radii | $(\rho_{\text{low}}, \rho_{\text{high}}) = (0.2, 0.5)$ |
| Band weight $\lambda_{\text{band}}$ | 0.1 |
| Reward tolerance $\varepsilon_r$ | adaptive ($0.1 \times$ std of rewards) |
| Acceptance gate | $|\hat{r}(s_c, a, z) - r| \leq \varepsilon_r, \|\Delta_t\|_2 \leq \rho_{\text{high}}$ |
| Training subset | Top-70% reward transitions |

## F.3. Hybrid Causal Policy Architecture

**Tokenization and embeddings** Trajectories are tokenized as repeating triplets $(\hat{G}_t, s_t, a_t)$ with context length $K = 20$ for MuJoCo and $K = 50$ for AntMaze. We use learned positional embeddings and layer normalization. The model dimension is $d_{\text{model}}=128$ with 2 heads (per-head dimension 64) and 1 transformer layer.

**Local processing** We build three parallel streams (RTG, state, action). Each stream applies depthwise 1D convolutions followed by pointwise projections with residual connections (kernel size 3, stride 1). Outputs are concatenated along the channel axis and projected back to $d_{\text{model}}$.

**Global reasoning** A single multi-head self-attention block operates on the fused sequence. We supervise the final attention block, with all heads constrained to place probability mass only on the direct parents of $a_t$, namely $s_t$ and $\hat{G}_t$, and optionally a short window of recent states. Let $A$ be the attention matrix and $q_{i,\cdot}$ the target distribution over valid parents $S_i$. The mask loss is

$$\mathcal{L}_{\text{mask}} = \frac{1}{L} \sum_{i=1}^{L} \sum_{j \in S_i} -q_{i,j} \log(A_{i,j} + \epsilon), \quad \epsilon = 10^{-8}.$$

The total loss is

$$\mathcal{L}_{\text{total}} = \mathcal{L}_{\text{action}} + \lambda_{\text{mask}} \mathcal{L}_{\text{mask}}.$$

*Table 7.* Hybrid causal policy architecture.

| Component | Setting | Notes |
|---|---|---|
| Streams | RTG / State / Action | Parallel, modality-specific |
| Conv blocks per stream | 2 | Depthwise $3\times1$ + pointwise $1\times1$ |
| Fusion | Concat + Linear | To $d_{\text{model}}$ |
| $d_{\text{model}}$ | 128 | Embedding size |
| Transformer layers | 1 (final only) | On top of conv fusion |
| Attention heads | 2 | Matches implementation |
| Supervised heads | All heads | Last block only |
| Supervise window $w$ | 0–4 | Default $w=2$ (recent states) |
| Decay for $s_{t-k}$ | $0.6^k$ | Geometric weights |
| Parent tokens | $s_t, \hat{G}_t \,(+\, s_{t-k})$ | Small weight on $a_{t-1}$ optional |
| $\epsilon$ in loss | $10^{-8}$ | Numerical stability |

**Policy training protocol.** We use Adam, batch size 64, weight decay $10^{-4}$, dropout 0.1, GELU activations, and a linear warmup schedule. Evaluation follows D4RL. Model selection uses validation return.

*Table 8.* Training configuration of CSET policy.

| Hyperparameter | Value |
|---|---|
| Optimizer | Adam |
| Learning rate | $1\times10^{-4}$ |
| Weight decay | $1\times10^{-4}$ |
| Batch size | 64 |
| Dropout | 0.1 |
| Activation | GELU |
| Context length $K$ (MuJoCo / AntMaze) | 20 / 50 |
| Learning rate schedule | Linear warmup |

### F.4. Details for Spurious Correlation Experiment

This appendix describes how we construct the spurious datasets and the exact evaluation protocol used in the robustness experiment.

**Dataset construction.** We create spurious variants of HalfCheetah-medium-v2, Hopper-medium-v2, and Walker2d-medium-v2 (D4RL) as follows: (1) Load the original offline trajectories. (2) Compute the global median of per-step rewards. (3) Define a binary distractor feature $d_t := \mathbb{I}\{r_t > \text{median}(r)\}$. (4) Append $d_t$ to each state vector (state dimension increases by one). (5) Save the modified dataset. This makes $d_t$ highly predictive of reward in the training data while having no causal effect on the environment.

**Training protocol.** DT and DC are trained directly on the modified datasets (with $d_t$ appended to states). For CSET, we first apply counterfactual augmentation to the same spurious dataset: the Causal Reward Model (CRM) and Counterfactual State Generator (CSG) generate candidate counterfactual states that are then filtered by the acceptance gates (reward consistency and move-band checks). The distractor $d_t$ is treated as part of the state input but is not used in abduction beyond its presence in $s_t$ and does not enter the reward-preservation objective directly. Accepted counterfactuals are merged with the spurious dataset to form the augmented training set.

**Evaluation protocol.** At test time, we break the learned correlation by a direct intervention in the evaluation loop. Policies are evaluated in the original, unmodified environments. For each observed state, we append a fixed distractor value of 0.0 before passing it to the policy, matching the training input dimension but removing any predictive content from the

distractor.[2]

**Robustness to Stochastic Distractors**   In our main robustness experiment (Section 4.3), we utilized a deterministic distractor $d_t = \mathbb{I}[r_t > \mathrm{median}(r)]$, which creates a perfect correlation ($R = 1.0$) between the feature and the reward class. To further validate our method under noisier spurious signals that mimic real-world sensor noise, we evaluated all methods on a stochastic distractor scenario.

We define a noisy distractor with a flip probability $p = 0.15$:

$$d_t = \mathbb{I}[r_t > \mathrm{median}(r)] \oplus \mathrm{Bernoulli}(0.15) \tag{6}$$

This distractor preserves the reward-aligned bit with probability 0.85 and randomly flips it with probability 0.15, reducing its reliability while keeping a noticeable correlation with reward. We trained DT, DC, CSET, ReBRAC, and EDAC on this stochastic dataset and evaluated them under the same intervention protocol (fixing $d_t = 0$ at test time).

Figure 2 shows the expert-normalized returns for all five methods under three settings: no distractor, the deterministic distractor, and the stochastic distractor. Adding noise to the spurious feature weakens its predictive value, and across all baselines the degradation is correspondingly smaller than in the deterministic case. This occurs because the stochastic distractor is a "noisier" predictor than the deterministic one; consequently, the models rely on it slightly less during training, leading to a smaller collapse when the feature is removed. However, the baselines' performance remains significantly degraded compared to the standard setting, confirming that they still suffer from causal confusion. In contrast, CSET demonstrates remarkable stability, achieving returns comparable to the clean baseline. These empirical results confirm that CSET's robustness benefits are not limited to deterministic artifacts but extend to stochastic correlations.

## G. Details on the proposed method

**Counterfactual Construction.**   Counterfactual Construction follows the Pearl's three-step causal procedure: abduction, action, and prediction (Pearl et al., 2000; Pearl, 2009).

1. Abduction: Infer the latent disturbance $z_t \sim q_\phi(z \mid s_t, a_t, r_t)$, which captures the unobserved factors that, together with $(s_t, a_t)$, produced the factual reward.

2. Action (Intervention): Define the counterfactual intervention by holding the action $a_t$ fixed. In our reward SCM, $a_t$ is a direct causal parent of $r_t$. To ensure the counterfactual is valid with respect to the reward mechanism, we hold the action parent fixed and solve for a state edit.

3. Prediction: Compute the counterfactual state $s_c = \mathrm{CSG}(s_t, a_t, z_t)$. This step generates the value of the state variable in the counterfactual world, ensuring it remains on-manifold and preserves the reward under the fixed action and abduced context.

In this procedure, $s_t$ provides the factual state as the basis for editing (prediction step), $a_t$ ensures the counterfactual remains consistent with the reward mechanism (intervention step), and $z_t$ ensures the generator respects the unobserved disturbance inferred from evidence (abduction step). Together, these allow CSET to follow the full abduction–action–prediction pipeline and generate valid counterfactual states.

**On the move-band constraint.**   The move band $[\rho_{\mathrm{low}}, \rho_{\mathrm{high}}]$ in Equation (3) serves two purposes beyond reward preserva-tion: (i) *avoiding trivial augmentations* — without a lower bound, the generator can collapse to near-identity edits that add little diversity; and (ii) *preventing off-manifold edits* — without an upper bound, large moves can drift outside dataset support and destabilize policy learning. We normalize state coordinates and apply the constraint in normalized space. Empirically, $(\rho_{\mathrm{low}}, \rho_{\mathrm{high}}) = (0.2, 0.5)$ yields non-trivial yet plausible edits.

**Acceptance gates at generation time.**   After training, we generate counterfactuals and accept $s_c$ only if

$$\left| \hat{f}_r(s_c, a_t, z_t) - r_t \right| \leq \varepsilon_r \quad \text{and} \quad \left\| \frac{s_c - s_t}{\sigma_s} \right\|_2 \leq \rho_{\mathrm{high}}.$$

---

[2]We use the same feature normalization as in training; the added distractor coordinate is standardized with the training statistics and set to the normalized value corresponding to $d_t{=}0$.

The tolerance $\varepsilon_r$ is set adaptively to $0.1 \times \text{std}(r)$ if not specified; only accepted $s_c$ are written back.

**Data augmentation procedure** For each factual trajectory $\tau = \{(s_t, a_t, r_t)\}_{t=0}^{T-1}$, we create a small number of *augmented copies*. In each copy, we scan time steps that pass a reward quantile filter and propose $s_c = g_\psi(s_t, a_t, \mu_\phi(s_t, a_t, r_t))$. If the candidate passes the two acceptance gates above, we replace only the observation token $s_t$ with the counterfactual $s_c$. The next observation $s_{t+1}$ is kept factual; we do not treat it as the successor of $s_c$. The final dataset on disk is the union of all original trajectories and their edited copies; no in-batch replacement is performed later.

# H. Ablation Studies on Counterfactual Generation

We conduct ablations to evaluate the design choices in the counterfactual generation pipeline. Experiments are performed on `hopper-medium-v2` and `halfcheetah-medium-v2`, with results reported as expert-normalized return (mean $\pm$ std over 5 seeds).

## H.1. Reward Consistency Gate

The counterfactual generator accepts a proposal only if

$$\left| \hat{f}_r(s_c, a_t, z_t) - r_t \right| \leq \varepsilon_r,$$

which ensures counterfactuals remain consistent with the factual reward. Without this gate, counterfactuals may provide contradictory training signals. In implementation, $\varepsilon_r$ is automatically set to $0.1 \times \text{std}(r)$, scaling the tolerance relative to reward variability in the dataset.

*Table 9.* Ablation on reward consistency gate. Expert-normalized returns (mean $\pm$ std over 5 seeds).

| Method | Hopper-medium | HalfCheetah-medium |
|---|---|---|
| CSET (full) | $93.4 \pm 1.5$ | $44.9 \pm 0.7$ |
| w/o reward gate | $74.7 \pm 3.3$ | $42.1 \pm 1.1$ |

We find that removing the gate significantly degrades Hopper performance, with a smaller but consistent drop on HalfCheetah. This confirms that label-preservation under the Causal Reward Model is critical for stable counterfactual training.

## H.2. Band Constraint

The move band constrains the normalized edit magnitude $\|\Delta_t\|_2 = \|(s_c - s_t)/\sigma_s\|_2$ to lie within $[\rho_{\text{low}}, \rho_{\text{high}}]$. This discourages both trivial near-copies ($\|\Delta_t\|_2 < \rho_{\text{low}}$) and implausible off-manifold shifts ($\|\Delta_t\|_2 > \rho_{\text{high}}$). We default to $(\rho_{\text{low}}, \rho_{\text{high}}) = (0.2, 0.5)$ and weight the penalty with $\lambda_{\text{band}} = 0.1$.

Table 10 shows that this constraint is critical for performance. Removing the band constraint entirely ($\lambda_{\text{band}} = 0$) causes a significant performance collapse on both environments. Relaxing only the lower bound ($\rho_{\text{low}} = 0$), which allows for trivial copies, also degrades performance, confirming the value of encouraging meaningful state edits.

*Table 10.* Ablation on the band constraint. Expert-normalized returns (mean $\pm$ std over 5 seeds).

| Method | Hopper-medium | HalfCheetah-medium |
|---|---|---|
| CSET (full) | $93.4 \pm 1.5$ | $44.9 \pm 0.7$ |
| w/o band constraint ($\lambda_{\text{band}} = 0$) | $68.2 \pm 4.1$ | $41.9 \pm 1.8$ |
| w/o lower band ($\rho_{\text{low}} = 0$) | $85.9 \pm 2.4$ | $43.7 \pm 0.9$ |

The diagnostics in Figure 5 and Figure 6 explain these choices.

**Move histograms.** The empirical $\|\Delta\|_2$ distributions are unimodal, with modes around 0.33–0.37. This suggests that the generator naturally prefers moderate edits rather than extremely small or large ones. Our default band of $(0.2, 0.5)$ brackets this region, retaining the majority of plausible moves while excluding both trivial near-copies ($< 0.2$) and overly aggressive

shifts ($> 0.5$). This balance prevents the generator from collapsing to identity mappings while also discouraging unrealistic counterfactuals.

**Acceptance heatmaps.** We further sweep ($\rho_{\text{low}}, \rho_{\text{high}}$) and report acceptance rates. The heatmaps show that our chosen range $(0.2, 0.5)$ maintains high acceptance (around 90%–95% on Hopper-medium and around 70%–75% for HalfCheetah-medium) while keeping edits within a safe move region. Tightening the lower bound (e.g., $\rho_{\text{low}} = 0.3$) sharply decreases acceptance, discarding many otherwise valid counterfactuals. Conversely, relaxing the upper bound (e.g., $\rho_{\text{high}} > 0.6$) increases acceptance but allows off-manifold edits (because the $\rho_{\text{high}}$ term is used in the acceptance gate).

**PCA overlays.** To assess the geometric plausibility of generated states, we visualize original and counterfactual states in the top two PCA components. Counterfactuals (red) largely overlap with the support of the original dataset (blue), indicating that edits remain on-manifold under our chosen range $(0.2, 0.5)$. The necessity of the band constraint and the reward consistency gate is paramount to achieving this result; ablation studies confirm that relaxing these constraints leads to off-manifold drift and subsequent degradation in policy performance. PCA preserves global variance and thus directly reflects whether counterfactual states share the dominant directions of variation with the original data; we treat it as the primary evidence for the on-manifold claim.

**t-SNE overlays.** As a complementary nonlinear projection, we also visualize original and counterfactual states under t-SNE, which preserves local neighborhood structure rather than global geometry. The picture is consistent with PCA: counterfactual states (red) interleave with original states (blue) across clusters rather than forming isolated regions. The agreement of two projection methods with different preservation properties (global linear variance for PCA, local nonlinear neighborhoods for t-SNE) strengthens the on-manifold claim beyond what either visualization alone can support.

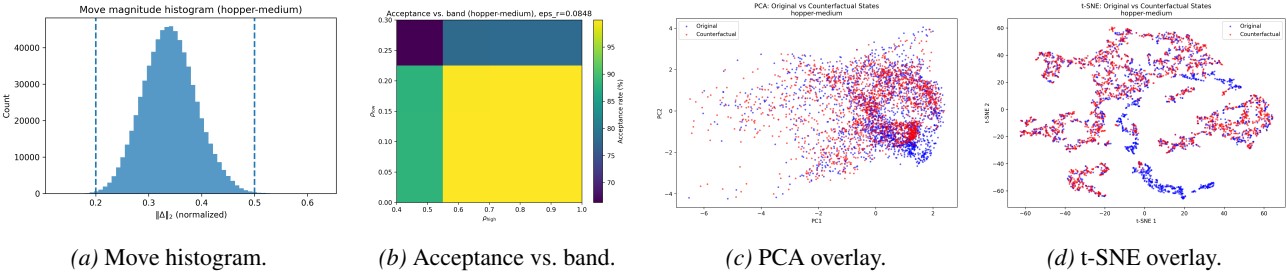

| (a) Move histogram. | (b) Acceptance vs. band. | (c) PCA overlay. | (d) t-SNE overlay. |

*Figure 5.* **Hopper-medium** diagnostics. The default band $(0.2, 0.5)$ achieves high acceptance while counterfactuals stay on-manifold under both PCA (global variance) and t-SNE (local neighborhood) projections.

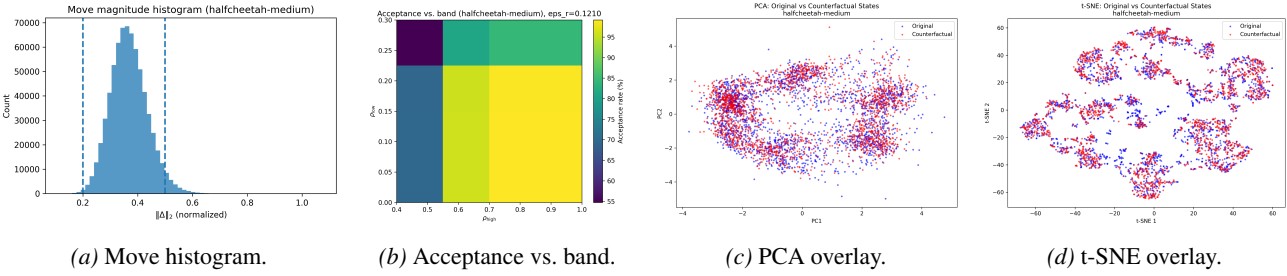

| (a) Move histogram. | (b) Acceptance vs. band. | (c) PCA overlay. | (d) t-SNE overlay. |

*Figure 6.* **HalfCheetah-medium** diagnostics. The default band $(0.2, 0.5)$ balances coverage and precision; counterfactuals remain aligned with the original manifold under both PCA and t-SNE projections.

## I. Short-Context Decision Transformer Does Not Address Causal Confusion

Several recent works have tested whether reducing the Decision Transformer (DT) context length improves stability. This idea was evaluated directly by the Decision ConvFormer (Kim et al., 2024). In their Appendix G.3 (Table 19), they report that DT performance decreases as the context window is shortened. These findings are consistent with our robustness results in Figure 2. Our experiments show that the standard DT fails because it overfits to spurious correlations, a problem of causal confusion that is not resolved by simply reducing the context length.

