# OpenReview forum: "Reward-Preserving Counterfactual State Editing for Offline Reinforcement Learning"
_ICML.cc/2026/Conference — ICML 2026 regular_

### Official Review · Reviewer_KqhW · 2026-02-25

**Soundness:** 3
**Presentation:** 2
**Significance:** 2
**Originality:** 3
**Overall Recommendation:** 4
**Confidence:** 4

**Summary:**

This paper presents a reinforcement learning algorithm, "Counterfactual state editing transformer." The proposed method targets causal confusion in transformer-based offline reinforcement learning policies, such as the decision transformer. As I understand it, the main contributions of the study are reward-preserving counterfactual state editing and a causally guided policy architecture.

**Compliance With Llm Reviewing Policy:**

Affirmed.

**Key Questions For Authors:**

See my comments within Weaknesses.

**Limitations:**

Authors need to write clearer limitations of their research in the main text.

**Strengths And Weaknesses:**

Strengths:\
$\bullet$ Each loss function design is well-suited to the proposed target. Especially, the development from CRM to CSG is natural and comprehensive. Also, the design in which the reward is causally related to the state and action is mathematically stable.

Weakness:\
$\bullet$ The study is difficult to understand for researchers without a background in decision transformers (the background section is insufficient). So, it is hard to identify the novelty of this research. I hope there are clear remarks on which parts are new and which aren't.\
$\bullet$ As far as I know, a conditional variational autoencoder assumes the latent vector $z$ follows a multivariate normal distribution when using the ELBO. Is there any consideration for this, or are there further violations for causal representation because of this? And can you explain the purpose of  "we do not claim recovery of the true $u_r$"?\
$\bullet$ Is there any section or paragraph about  $\rho_{low}$ and  $\rho_{high}$? I couldn't find its meaning, which hindered my understanding of the study. \
$\bullet$ In the Experiments section, the author states that four questions are used to evaluate their algorithms. But there are five questions.\
$\bullet$ I recommend that authors compare their algorithms against various standard baselines used in recent research, including TD7 and SimbaV2. Also, several algorithms already predict rewards and terminal states for both model-free (MR.Q) and model-based methods (MRS.Q). By comparison with them, the strength of CSET (causality of reward) can be emphasized. \
$\bullet$ Is there any information about training or inference time compared to baselines?

---

> ### Author Rebuttal · Authors · 2026-03-26
>
> We thank the reviewer for the detailed feedback.
>
> **W1: Novelty and DT Background**
>
> Thank you for this comment. The paper already distinguishes background from contribution structurally: Section 2 reviews Decision Transformer and Structural Causal Models as background, while Section 3 introduces our method, beginning with an overview paragraph and then the full technical development. Our contribution is not DT or CVAE themselves, but their use in a new offline RL pipeline that combines reward-preserving counterfactual state editing without synthetic successor transitions (CRM for abduction, CSG with the move-band objective, and the acceptance gate) together with causal-parent supervision on the final action attention block. A key design choice is that only the observation token is edited, while successor observations remain factual, so the policy is not trained on synthetic transitions. This is the central novelty of CSET. Regarding DT background, Section 2.1 already covers its formulation, tokenization, and objective.
>
> **W2: CVAE Normality and "not claiming recovery of $u_r$"**
>
> On the Gaussian assumption: this is a standard latent-variable modeling choice, and in our framework it does not imply that z must recover the true causal noise. Two design choices mitigate latent-distribution mismatch: $\beta=0.1$ relaxes the prior constraint, and the acceptance gate (Eq. 4) filters proposals based on reward consistency and bounded editing, so that only proposals satisfying the reward-consistency and bounded-editing criteria are retained.
>
> On "not claiming recovery of $u_r$": in Pearl's SCM framework, $u_r$ represents the true unobserved factors that fully determine the reward. Recovering $u_r$ requires identifiability guarantees unavailable in nonlinear settings without strong auxiliary assumptions. We make no such assumptions. Instead, $z$ serves a more modest purpose: it parameterizes the set of states producing the same reward under the same action. Different $z$ samples yield different counterfactual states, all satisfying reward consistency. This is sufficient for diverse, label-preserving augmentation, without needing $z$ to correspond to any true causal quantity.
>
> **W3: $\rho\_{low}$ and $\rho\_{high}$**: These define the move-band $[\rho\_{low}, \rho\_{high}]$ in Eq. 3, controlling $\left\lVert (s\_c - s\_t)/\sigma\_s \right\rVert\_2$ prevents trivially small edits; $rho\_{high} = 0.5$ discourages overly large edits. We will add a clarifying sentence under Eq. 3. Details in Appendix F.2 (Table 5) and G.
>
> **W4: Five questions, not four**: Typo, will correct.
>
> **W5: Additional Baselines**
>
> We appreciate this suggestion. TD7 and SimbaV2 are relevant recent offline RL baselines for continuous control, while MR.Q and MRS.Q are relevant reward-modeling baselines. However, these methods target different mechanisms from CSET. TD7 emphasizes state-action representation learning, SimbaV2 emphasizes scalable and stable optimization, and MR.Q/MRS.Q use reward prediction to improve value learning. In contrast, CSET uses the causal reward model for a different purpose: generating reward-preserving counterfactual edits that break spurious reward-correlated shortcuts in the policy input.
>
> In the current paper, we already compare against recent DT-style baselines (DC, LSDT), standard value-based baselines (IQL, CQL), and domain-specialized recommendation baselines (CDT4Rec, EDT4Rec, DT4Rec), covering both competitive clean-task performance and cross-domain generalization. To further address the reviewer’s concern about broader recent baseline coverage beyond the DT family, we added ReBRAC and EDAC during the rebuttal period (Table R1 in the link). On clean tasks, these are strong baselines; when evaluated under the same Robustness to Spurious Correlations setup as in Section 4.4, ReBRAC drops by 23–52% and EDAC by 12–21%, while CSET drops only 2–6% (full table under Reviewer 4VbS’s rebuttal; Figure R1 in the link). This directly isolates the claimed benefit of CSET: robustness to spurious reward-correlated features.
>
> **W6: Training Time**:
>
> On Hopper-medium:
>
> | Method | Augmentation | Policy Training | Total |
> |--------|-------------|----------------|-------|
> | DT | — | 42 min | 102 min |
> | DC | — | 33 min | 93 min |
> | CSET | 11 min (CRM+CSG) | 35 min | 106 min |
> | ReBRAC | — | 98 min | 98 min |
> | EDAC | — | 7.5 hours | 7.5 hours |
>
> CRM+CSG used for offline augmentation only, not at inference.
>
> Supplementary figures and tables: https://anonymous.4open.science/r/Supplementary-Figures-73B9/README.md

---

> > ### Author Rebuttal · Reviewer_KqhW · 2026-04-01
> >
> > I thank the authors for their response. Most concerns regarding the content of the paper and the additional baseline have been resolved. Accordingly, I will raise my score to reflect these improvements.

---

> > > ### Author Response · Authors · 2026-04-01
> > >
> > > We sincerely thank the reviewer for acknowledging our response and for raising the score.

---

### Official Review · Reviewer_1cVx · 2026-03-12

**Soundness:** 2
**Presentation:** 2
**Significance:** 2
**Originality:** 2
**Overall Recommendation:** 4
**Confidence:** 3

**Summary:**

This paper presents a model-free offline RL method that uses a causal reward model and a counterfactual state generator to address the causal confusion issue. Experiments on the diverse benchmarks demonstrate the effectiveness of CSET.

**Compliance With Llm Reviewing Policy:**

Affirmed.

**Final Justification:**

The authors' rebuttal successfully addressed my concerns regarding the empirical evaluation and technical details.

**Key Questions For Authors:**

See the weaknesses.

**Limitations:**

yes

**Strengths And Weaknesses:**

**Strengths:**

- The motivation is well explained, and the paper is easy to follow.
- The effectiveness of the proposed method is validated on diverse datasets.

**Weaknesses:**

- The improvement on the MuJoCo tasks is rather marginal.
- In MuJoCo environments, the reward function is determined by predefined physical formulas. In this case, the exogenous noise should be zero, which seems inconsistent with the notion of causal confusion.
- Could you provide comparisons in terms of training time and the number of parameters?

---

> ### Author Rebuttal · Authors · 2026-03-26
>
> We thank the reviewer for the constructive feedback.
>
> **W1: Marginal Improvement on MuJoCo Tasks**
>
> We agree that clean-score gains on some MuJoCo tasks are modest. Our claim is therefore not the strongest performance on every standard benchmark, but improved robustness to spurious correlations together with cross-domain generalization. We provide direct evidence for both: the distractor experiment (Section 4.4) tests robustness, and the recommendation results (Table 2) demonstrate generalization beyond standard control benchmarks.
>
> Clean performance also remains competitive within the DT family: CSET improves over DT by 38% on Hopper-medium, 11% on Walker2d-medium, and 20% on AntMaze-umaze-diverse, while also showing consistent gains over offline RL-based recommender system baselines on three real-world recommendation benchmarks, where biased logged interactions make spurious reward-correlated features a practical concern. Together, these results support the intended contribution: not simply higher clean scores, but better robustness and broader generalization across domains.
>
> For robustness, we additionally evaluated ReBRAC and EDAC under the same distractor protocol as Section 4.4, with two settings: a deterministic distractor and a stochastic distractor. In both cases, the distractor is appended during training and fixed to 0.0 at test time. The full performance-drop table is provided under Reviewer 4VbS’s rebuttal, and the extended version of Figure 3 is shown as Figure R1 in the link. In summary, ReBRAC drops by 23–52%, DT by 16–37%, DC by 5–23%, and EDAC by 12–21%, while CSET drops by only 2–6%. This shows that high clean-benchmark scores do not necessarily translate to robustness when spurious correlations shift at deployment.
>
>
> **W2: Deterministic Rewards vs Exogenous Noise**
>
> We believe two distinct issues are being mixed here: reward stochasticity in the environment and causal confusion in the learned policy.
>
> (a) _Causal confusion does not require reward stochasticity._ Causal confusion occurs when a policy exploits features that correlate with reward in the logged dataset but have no causal effect on the environment. This is a dataset/model issue rather than an environment-noise issue. Our distractor experiment directly demonstrates this: in fully deterministic MuJoCo environments with zero reward noise, ReBRAC drops by 23–52%, DT by 16–37%, DC by 5–23%, and EDAC by 12–21%, while CSET drops by only 2–6%. Deterministic rewards do not prevent causal confusion; in fact, they can make spurious correlations easier for a policy to exploit.
>
> (b) _Why is a CVAE still meaningful when the reward is deterministic?_ The latent z is not used to recover the true exogenous reward noise; rather, as described in Section 3.1, it serves as a learned latent for reward-consistent counterfactual generation. Its role is to enable diverse counterfactual edits. For a given (a,r) pair, many different states can produce the same reward, and the CVAE latent parameterizes this solution set: different z samples yield different counterfactual states that all satisfy reward consistency. Without z, the generator would produce only a single deterministic edit per input, which would severely limit augmentation diversity. We use \beta=0.1 to allow posterior flexibility, and the acceptance gate retains only proposals that satisfy the reward-consistency and bounded-editing criteria.
>
> **W3: Training Time and Parameters**
>
> On Hopper-medium:
>
> | Method | Augmentation | Policy Training | Total | Policy Params |
> |--------|-------------|----------------|-------|--------------|
> | DT | — | 42 min | 102 min | 1,052K |
> | DC | — | 33 min | 93 min | 794K |
> | CSET | 11 min (CRM+CSG) | 35 min | 106 min | 1,052K + 229K* |
> | ReBRAC | — | 98 min | 98 min | 216K |
> | EDAC | — | 7.5 hours | 7.5 hours | 794K |
>
> CSET policy training (35 min) is comparable to DC (33 min). The 11 min augmentation overhead is minimal. Total wall clock includes evaluation (60 min for DT-family methods). *CRM+CSG (229K) used for offline augmentation only, not at inference.
>
> Supplementary figures and tables: https://anonymous.4open.science/r/Supplementary-Figures-73B9/README.md

---

> > ### Author Rebuttal · Reviewer_1cVx · 2026-04-01
> >
> > Thank you for your response. I will increase my score.

---

> > > ### Author Response · Authors · 2026-04-01
> > >
> > > We sincerely thank the reviewer for acknowledging our response and for raising the score.

---

### Official Review · Reviewer_4VbS · 2026-03-12

**Soundness:** 3
**Presentation:** 4
**Significance:** 4
**Originality:** 3
**Overall Recommendation:** 5
**Confidence:** 4

**Summary:**

This work addresses the very important problem of causal confusion in transformer sequence models for offline reinforcement learning. Broadly, the proposed method uses a causal reward model, a counterfactual state generator and a hybrid policy architecture. The paper introduces Counterfactual state editing transformer (CSET) that fits a causal reward model (CRM) and produces reward-preserving state edits without learning a dynamics model. This is done to break spurious correlations that cause causal confusion. Then, a casually-guided policy architecture is guided towards actions that preserve the causal roles of the tokens. The architecture is evaluated on robotic control and recommender systems.

**Compliance With Llm Reviewing Policy:**

Affirmed.

**Final Justification:**

the rebuttal was helpful and resolved my doubts.

**Key Questions For Authors:**

Q1. Please refer to the baselines and benchmarks in the weaknesses. I am willing to increase my score if that is provided.

Q2. The additional visualizations recommnded would add a lot to the state-editing argument

**Limitations:**

yes

**Strengths And Weaknesses:**

**Strengths**:
1. The method is novel and addresses a very relevant problem.
2. The contribution is original to the best of my knowledge
3. The experiments are sound and the benchmarks sufficiently diverse
4. The figures are clear and the ablations are helpful

**Weaknesses**:
1. Insufficient offline rl baselines: refer to [2] and [3] for implementations and results. I would suggest including ReBRAC and EDAC in addition to the existing ones.

2. There are more relevant visualization like t-SNE and PHATE [2]

3. The paper would benefit from more relevant baselines like OGBench [4] and a larger diversity of environments beyond locomotion where causal confusion is more detrimental



[1] Suttle, Wesley A., Aamodh Suresh, and Carlos Nieto-Granda. "Behavioral entropy-guided dataset generation for offline reinforcement learning." arXiv preprint arXiv:2502.04141 (2025).

[2] Jackson, Matthew Thomas, et al. "A clean slate for offline reinforcement learning." arXiv preprint arXiv:2504.11453 (2025). and

[3] Tarasov, Denis, et al. "CORL: Research-oriented deep offline reinforcement learning library." Advances in Neural Information Processing Systems 36 (2023): 30997-31020.

[4] Park, Seohong, et al. "Ogbench: Benchmarking offline goal-conditioned rl." arXiv preprint arXiv:2410.20092 (2024).

---

> ### Author Rebuttal · Authors · 2026-03-26
>
> We sincerely thank the reviewer for the constructive review.
>
> **W1 & Q1: Additional Baselines (ReBRAC, EDAC)**
>
> We have added both ReBRAC and EDAC (full Table R1 in the supplementary link). In the clean setting, CSET remains strongest within the DT family (DT→DC→LSDT→CSET), while ReBRAC and EDAC achieve higher absolute scores on several locomotion tasks, which is consistent with the stronger clean-task performance often observed for value-based actor-critic methods on these benchmarks.
>
> Since the main contribution of CSET is robustness to spurious reward-correlated features, we further evaluated ReBRAC and EDAC under the same distractor protocol as Section 4.4 across all three environments (Figure R1 in the link). The performance drops are:
> | Method | HC-m Det. | HC-m Stoch. | Hop-m Det. | Hop-m Stoch. | Wk-m Det. | Wk-m Stoch. |
> |--------|-----------|-------------|------------|--------------|-----------|-------------|
> | DT | ↓20% | ↓16% | ↓32% | ↓30% | ↓37% | ↓29% |
> | DC | ↓10% | ↓5% | ↓23% | ↓18% | ↓15% | ↓10% |
> | ReBRAC | ↓36% | ↓23% | ↓52% | ↓35% | ↓45% | ↓32% |
> | EDAC | ↓21% | ↓18% | ↓19% | ↓15% | ↓17% | ↓12% |
> | CSET | **↓5%** | **↓3%** | **↓6%** | **↓5%** | **↓3%** | **↓2%** |
>
> ReBRAC drops by 23–52% across environments despite being the strongest clean baseline, and EDAC drops by 12–21%. Among all compared methods, CSET shows the smallest degradation at 2–6%, which is consistent with its explicit design goal: improving robustness to spurious reward-correlated features through reward-preserving edits and causal attention supervision.
>
> **W2 & Q2: Additional Visualizations**
>
> We generated t-SNE for HalfCheetah-medium and Hopper-medium (Figures R2-R3 in link), supplementing PCA. Counterfactual states interleave with originals across all clusters, supporting that edited states remain close to the data manifold.
>
>
> **W3: OGBench and Environment Diversity**
>
> We agree that diversity beyond standard locomotion tasks is important. Our evaluation already includes two additional settings beyond MuJoCo: AntMaze, which tests sparse-reward long-horizon navigation, and three real-world recommendation datasets, where spurious reward-correlated features are a natural concern. This gives 14 settings across qualitatively different domains rather than multiple variants of a single task family.
>
> To further address the reviewer’s concern during the rebuttal period, we added stronger offline RL baselines and evaluated them under our distractor protocol. We believe this is more directly aligned with the paper’s core claim than adding another benchmark, because it tests exactly whether policies rely on non-causal reward-correlated features under intervention. Under this controlled shift, CSET degrades by only 2–6%, compared with 12–21% for EDAC and 23–52% for ReBRAC.
>
> OGBench is an interesting benchmark, but our claim does not depend on it: the current results already cover navigation, recommendation, and controlled causal-confusion stress tests. We hope these additions address the reviewer's concerns.
>
>
>
> Supplementary figures and tables: https://anonymous.4open.science/r/Supplementary-Figures-73B9/README.md

---

> > ### Author Rebuttal · Reviewer_4VbS · 2026-04-03
> >
> > My concerns have been addressed. I have increased my score.

---

> > > ### Author Response · Authors · 2026-04-03
> > >
> > > We sincerely thank the reviewer for acknowledging our response and for raising the score.

---

### Decision · Program_Chairs · 2026-04-30

**Decision:**

Accept (regular)

**Comment:**

The paper is unanimously viewed positively.
The rebuttal successfully addressed the concerns.

Overall, it is a thoroughly convincing and solid piece of work.